# Diauxie and co-utilization of carbon sources can coexist during bacterial growth in nutritionally complex environments

Elena Perrin [1,7], Veronica Ghini[2,7], Michele Giovannini[1], Francesca Di Patti[3,4], Barbara Cardazzo[5], Lisa Carraro[5], Camilla Fagorzi [1], Paola Turano[6], Renato Fani[1] & Marco Fondi [1,4✉]

It is commonly thought that when multiple carbon sources are available, bacteria metabolize them either sequentially (diauxic growth) or simultaneously (co-utilization). However, this view is mainly based on analyses in relatively simple laboratory settings. Here we show that a heterotrophic marine bacterium, *Pseudoalteromonas haloplanktis*, can use both strategies simultaneously when multiple possible nutrients are provided in the same growth experiment. The order of nutrient uptake is partially determined by the biomass yield that can be achieved when the same compounds are provided as single carbon sources. Using transcriptomics and time-resolved intracellular $^1$H-$^{13}$C NMR, we reveal specific pathways for utilization of various amino acids. Finally, theoretical modelling indicates that this metabolic phenotype, combining diauxie and co-utilization of substrates, is compatible with a tight regulation that allows the modulation of assimilatory pathways.

[1] Department of Biology, University of Florence, Florence, Italy. [2] Consorzio Interuniversitario Risonanze Magnetiche di Metallo Proteine (CIRMMP), Florence, Italy. [3] Department of Physics and Astronomy, University of Florence, Florence, Italy. [4] CSDC, University of Florence, Florence, Italy. [5] Department of Comparative Biomedicine and Food Science, University of Padova, Padova, Italy. [6] Center of Magnetic Resonance (CERM), University of Florence, Florence, Italy. [7] These authors contributed equally: Elena Perrin, Veronica Ghini. ✉email: marco.fondi@unifi.it

Microorganisms must quickly and efficiently adapt to a variety of possible fluctuations in the surrounding environment. When considering changes in the pool of available nutrients, this is usually achieved by a tight regulation of their metabolic phenotypes by sensing the availability of specific compounds, synthesizing the enzymes required for their catabolism and repressing them after specific metabolites are depleted[1]. The spectrum of possible bacterial metabolic adaptation strategies can be observed, for example, when growing cells in a medium containing a simple mixture of carbon sources. In this situation bacterial may exhibit different patterns including diauxic growth[2], simultaneous consumption[3], and bistable growth[4,5]. Further, nutrients concentration and growth medium composition are known to affect other important cellular features such as motility and cell adhesion[6,7] and biofilm formation[8,9]. Typically, these phenomena have been studied (both theoretically and experimentally) in model organisms (e.g., *Escherichia coli* and *Lactococcus lactis*)[10], grown on defined media containing simple mixtures of 2/3 carbohydrates, e.g., glucose and lactose[11–14]. In natural conditions, however, bacteria rarely encounter simple combinations of exploitable carbon/energy sources. Rather, complex mixtures of nutrients are common and often colonized by actively growing bacteria. This is the case, for example, of intracellular pathogens that are commonly faced with a diverse set of host nutrients in infected tissues. In these cases, bacteria have been shown to adapt to this situation by the simultaneous exploitation of plastic and flexibles nutrient utilization strategies[15–19].

Do the same models developed for simplified conditions hold also in real-case scenarios? At present, we witness a knowledge gap concerning the study of these processes in experimental settings that do not involve model organisms and/or defined media and we lack a sound theoretical understanding of the mechanisms driving nutrients assimilation strategies in conditions that are closer to the ones found in natural settings.

Bacterial exploitation of nutrient patches is made up of (at least) two different stages, i.e., physical interaction followed by carbon sources metabolic degradation. The capability of bacteria to interact with transient nutrient sources is well documented and has revealed their high efficiency in exploiting transient nutrient patches[20,21]. Little is known, instead, on the molecular aspects regulating and influencing bacterial productivity once micro-scale nutrient hot spots are colonized. At this stage, i.e., when cells start to feed on the available carbon source(s), other cellular mechanisms need to be involved to ensure a systematic exploitation of the resource. Indeed, as nutrient patches are likely composed of complex nutrient mixtures (that may include carbohydrates, amino acids, lipids and nucleic acids) bacteria need to dynamically activate specific degradation pathways according to the kind and concentration of external nutrients. In other words, a continuous and flexible genetic reprogramming needs to be active to ensure that the preferred compound(s) are sequentially or simultaneously taken up from the external environment and properly metabolized. Up to now, this latter aspect has been mostly overlooked despite it might be central in the understanding of micro-scale nutrients dynamics.

In this regard, the marine environment represents a paradigmatic example of the challenges encountered by microorganisms when it comes to the efficient (and rapid) exploitation of complex nutritional inputs. Such a habitat is thought to be characterized by a low average nutrient level (e.g., the concentration of amino acids is in the range of $\sim 10^{-9}$ M) and nutrients in general appear and disappear in a sporadic fashion, demanding a precise chemical response, a fast swimming speed, and ability to localize and exploit a nutrient patch once it is found[6]. These are the conditions that are commonly faced by marine heterotrophic bacteria, i.e., those microorganisms relying on the assimilation of external biomass for both energy generation and nutrition. Their metabolism is pivotal for the maintenance and the correct balance of oceanic biogeochemical cycles as they are central to the so-called microbial loop, i.e., the trophic pathway of the marine food web responsible for the microbial assimilation of dissolved organic matter, by transforming phytoplankton-derived organic matter and fuelling the entire ocean biogeochemical nutrient cycle. In this regard, the metabolism of *Pseudoalteromonas haloplanktis* TAC125 (PhTAC125), a heterotrophic marine bacterium isolated from Antarctica, has recently gained a certain attention due to its potential biotechnological exploitation[22], its capability to synthesize anti-biofilm compounds[23], the necessity to set up efficient culture conditions[24,25] and the metabolic reprogramming during growth in complex environments[26]. In particular, the analysis of its growth phenotype in an amino acid rich medium has shown the presence of metabolic switches among different groups of amino acids[25], although nothing could be said about the molecular mechanisms underlying such phenotype and the possible regulation involved. Using constraint-based metabolic modelling we attempted to provide a systems level scheme of PhTAC125 metabolic re-wiring as a consequence of carbon source switching in such a nutritionally complex medium. Our simulations highlighted an efficient reprogramming of PhTAC125 metabolic machinery to quickly adapt to a nutritionally unstable environment, compatible with adaptation to fast growth in a highly competitive environment[26].

Here we have investigated the global regulation of a marine heterotrophic bacterium when grown in both a complex and a defined rich medium (i.e., including multiple possible carbon sources) using and integrating a set of complementary-omics techniques (i.e., transcriptomics and $^1$H and $^{13}$C NMR metabolomics) with measured growth parameters. We show that the two main nutritional strategies commonly observed (co-utilization and sequential uptake of multiple substrates) can coexist in the same growth experiment, leading to an efficient exploitation of the available carbon sources. We also developed two theoretical models accounting for nutrients switching in a nutritionally rich environment in the presence and absence of cell regulation acting at the level of resource allocation in the synthesis of nutrient assimilation pathways. We show that a model taking into consideration an overall regulatory control on the sequence of nutrients uptake produces a better fit with available experimental data with respect to a purely Michaelis–Menten kinetic model.

## Results

**Global regulation of a triauxic growth.** *P. haloplanktis* TAC125 (hereinafter PhTAC125) cells were grown in shaken flasks in a complex medium composed of Schatz salts[27] and peptone as their C source. Optical density (OD) was measured every hour and cellular RNA was sampled in five different time points of their growth (Fig. 1a). To increase the time points for a better growth rate estimation, we also used an interpolation technique on the data generated at this stage. Details on the specific interpolation approach used are reported in "Methods" and the resulting plot is reported in Supplementary Fig. 1. The growth curve displays a triauxic pattern (Fig. 1a). An initial growth phase (growth rate of 0.023 h$^{-1}$, 0.021 considering interpolated data) is interrupted by a lag phase between min. 180 and min. 240; afterwards, cells start growing over but such growth is interrupted by another lag phase between min. 280 and min. 340. Cells then started growing again, until the end of the experiment (growth rate of 0.004 h$^{-1}$, 0.006 h$^{-1}$ considering interpolated data). The average growth rate across all the time points was estimated to be 0.01 h$^{-1}$. To identify transcriptional changes during cell growth total RNA was extracted and

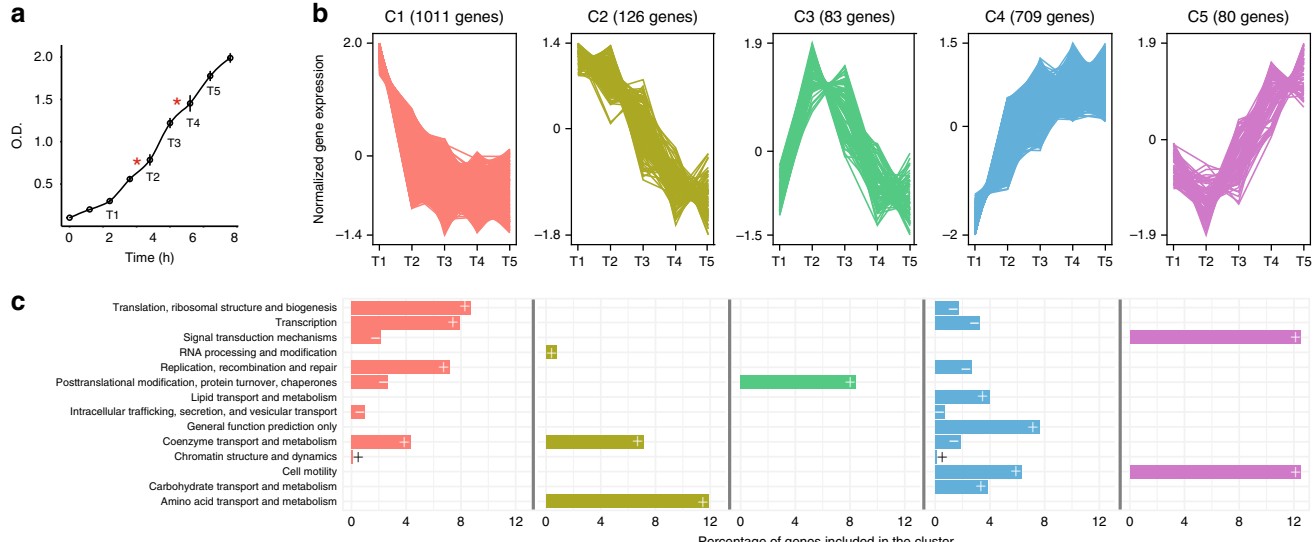

**Fig. 1 Transcriptomic data of the multi-auxic growth. a** PhTAC125 growth curves in peptone. Asterisks represent hypothetical switch points and T1–T5 indicate the sampled time points. **b** Clustering of PhTAC125 genes according to their expression values (normalized according to *clust* algorithm, see "Methods") throughout the growth curve. **c** COG functional categories that are significantly over- ("+" sign) or under- ("−" sign) represented inside each cluster. Source data are provided as a Source Data file.

sequenced (two biological and two technical replicates for a total of 20 samples) using Illumina MiSeq (Genomix4Life, Naples, Italy). The main features of the 20 sequenced samples are reported in Supplementary Table 1. We clustered the genes according to their expression during the growth and identified six major trends (clusters C1–C6, Fig. 1b). Overall, we were able to cluster 2045 genes out of the 3448 encoded by the PhTAC125 genome (roughly 60%). We then performed a functional annotation and a functional enrichment analysis for the genes embedded in each cluster. One of these clusters (C6) did not include any significantly enriched functional category and thus it was discarded. Cluster C1 includes genes that display a decrease in their expression between the first two time points (T1 and T2) and a constant (low) expression across the rest of the growth curve.

Over-represented genes embedded in this cluster included those involved in basic housekeeping functions such as translation, DNA replication and transcription (Fig. 1c). Genes embedded in Cluster C2 displayed a decreasing trend throughout the growth curve and mainly included genes involved in RNA processing, metabolism of coenzymes and amino acids transport and metabolism. The expression of the 83 genes included in Cluster C3 was characterized by an abrupt increase between T1 and T2 and then an overall decrease until the end of the curve. This cluster significantly included genes involved in post-translational modification, protein turnover and chaperons. Clusters C4 and C5 included genes whose expression tended to increase in the later stages of the growth; over-represented genes in C4 mainly belonged to lipid metabolism, cell motility and amino acids transport and metabolism. The expression of genes included in C5 decreased during the first stages of the growth and is then increased for the rest part of the curve. The cluster of genes included those involved in signal transduction mechanisms and cell motility.

Whole-genome transcriptomics data depict a scenario in which PhTAC125 is active and fast-growing mainly during the first stages of the curve, as reflected by the relatively high expression of translation, transcription replication and coenzyme metabolism genes. Genes embedded in these categories are under-represented among those increasing their expression in the last stages of the growth (Fig. 1c) and over-represented among those with high

expression values in the first stages of the growth. Metabolically, PhTAC125 cells seem to rely more on amino acids metabolism in the initial stages of their growth, consistently with their progressive exhaustion in the medium. The last part of the growth experiment was also characterized by an increase in gene expression of cell motility-related genes (over-represented in C4 and C5). Finally, genes generally related to post-processing mechanisms peak their expression at T2.

**A non-*E. coli*-like regulatory response to nutrients exhaustion.**
The triauxic growth curve reported in Fig. 1a (and in Supplementary Fig. 1 using interpolated data) suggests the presence of a dynamic control on the adjustment of cell physiology. Here we sought to quantify the regulatory effort required to growing cells for modulating such cellular response. We focused on transcriptional factors (TFs) and two-component response systems (TCRSs) and analyzed differentially expressed genes among three points of PhTAC125 growth curve, namely T1 vs. T3 and T3 vs. T5. These points should capture PhTAC125 cells during exponential growth after the first growth lag (T1), in-between the two growth lags (T3) and after the final growth lag but before getting to plateau (T5).

First, we checked whether PhTAC125 regulation system somehow resembled the model scheme of the known overall metabolic regulation (i.e., the one characterized in *E. coli*). Of the 81 transcription factors known to directly or indirectly control central metabolic enzymes[28], we found a reliable homologue ($E$-value < $1e^{-20}$) only for 34 of them (Supplementary Note 1 and Supplementary Data 1). PhTAC125, for example, lacks key players in bacterial diauxic shifts as the major global regulator of catabolite-sensitive operons (when complexed to cAMP) *crp* and the genes responsible for the synthesis of cyclic AMP (adenylate cyclase, *cyaA*). Among the 34 global regulators identified, only ten (roughly 25% of the shared ones and 12% of the entire *E. coli* set) displayed a significantly altered expression following the first transition (T1–T3) and none of them was differentially expressed following the second one (T3–T5). Details on the shared, differentially expressed TFs are provided in Supplementary Table 2.

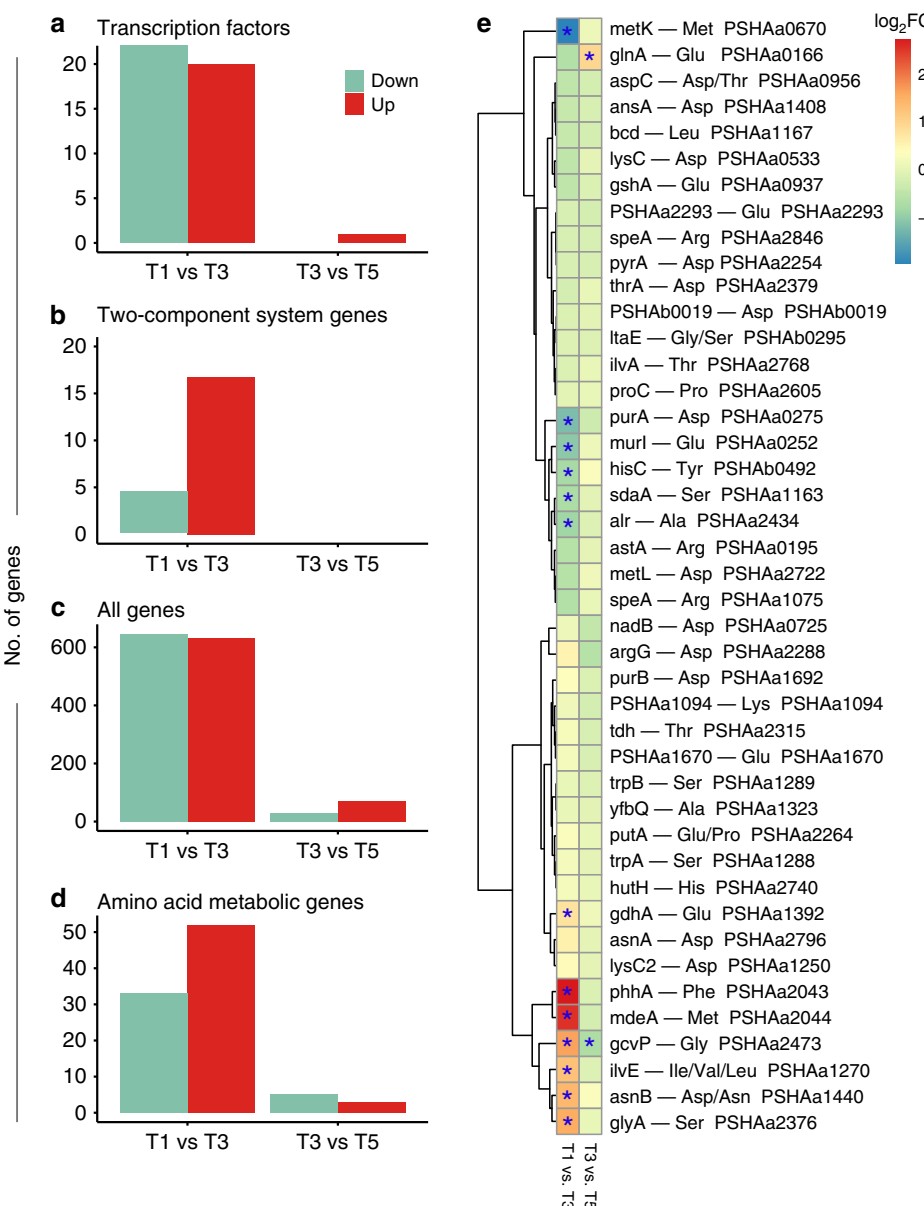

**Fig. 2 Differentially expressed genes. a–d** Number of up- and downregulated genes in the two contrasts considered, for different gene categories. **e** Fold change of each gene responsible for the first degradative step of each amino acid. Differentially expressed genes are marked with an asterisk. Source data are provided as a Source Data file.

A similar situation was observed for eight selected sigma-factors that control gene expression globally[28]. In this case, as expected, an ortholog was found for each of them, but only two of them showed an altered expression following T1–T3 transition and the expression of none of them was significantly altered following T3–T5 one. The two genes displaying a significant change in gene expression were *rpoS* and *rpoD*. RpoS is the primary regulator of stationary phase genes, whereas RpoD is the primary sigma factor during exponential growth. Expectedly, the first resulted to be upregulated following the T1–T3 transition, whereas the second was downregulated (Supplementary Table 3).

With the exception of RpoS and RpoD, whose expression is in line with the global control of exponential vs. stationary phases, it appears that growth lags are regulated by mechanisms that poorly overlap with our current knowledge.

For this reason, we evaluated the expression of the entire repertoire of PhTAC125 TFs across the two points that involved the ceasing of cellular growth in our experiment, i.e., T1–T3 and

T3–T5. Overall, we identified 41 differentially expressed TFs, 22 downregulated and 19 upregulated (Fig. 2b, Supplementary Fig. 2 and Supplementary Data 2) following the first growth interruption. The second growth lag was characterized by the significant change in expression (upregulation) of just one TF. Together with TFs, TCRSs are a basic stimulus-response coupling mechanism to sense and react to changes in environmental conditions, e.g., nutrient concentration. We identified differentially expressed TCRSs in the two selected contrasts. Overall, we found 21 TCRSs-related genes that were differentially expressed in T1 vs. T3 and none in the T3 vs. T5 transition (Fig. 2c and Supplementary Data 3).

We conclude that the two growth lags observed in the curve apparently point to different reprogramming efforts that, in turn, may underpin distinct nutrients uptake strategies. The first growth interruption seems to have a deeper impact on the entire metabolic system, whereas the second could imply a fine tuning of the catabolic machinery. This is further confirmed by the

overall number of DEGs across the selected contrasts (Fig. 2a), 1280 between T1 and T3 (633 and 647 up- and downregulated genes, respectively) and only 101 between T3 and T5.

**Amino acid assimilation pathways and their (dis)regulation**. Previous experiments have shown that PhTAC125 displays a coordinated sequence of amino acids degradation when grown in medium embedding complex mixtures of such molecules (i.e., peptone or casamino acid-based media, Supplementary Note 2)[25]. In other words, some amino acids are preferred over others and are metabolized early in PhTAC125 growth curve. This switching among nutrients suggests that an active and modulated reprogramming occurs during PhTAC125 growth in a nutritionally complex environment. Similarly, differentially expressed amino acid metabolic genes (hereinafter AA-genes) are unevenly distributed among the two contrasts considered (Fig. 2d). T1 vs. T3 displays a higher number of DEGs (85, 33 downregulated and 55 upregulated) with respect to T3 vs. T5 (8, 5 downregulated and 3 upregulated). Considering the amino acid assimilation pathways of differentially expressed AA-genes in the T1 vs. T3 contrast (Supplementary Fig. 3), we did not observe a clear functional bias towards specific routes. Almost all the pathways are represented, both in terms of up- and downregulated genes. Similarly, the switch between T3 and T5 included downregulated genes involved in a broad spectrum of metabolic pathways including Val, Leu and Ile degradation, Tyr metabolism and Ala, Asp and Glu metabolism (one gene for each pathway) and upregulated genes in Gly, Ser and Thr metabolism (1 gene), Lys biosynthesis (1 gene) and Arg biosynthesis (3 genes).

To unravel the faith of each amino acid inside the cell, we analyzed the expression of the genes involved in all their possible first assimilatory step (Fig. 2e and Supplementary Data 4). We found 13 DEGs in the T1–T3 contrasts and 2 DEGs in the T3–T5 contrast. The first set comprised seven upregulated and six downregulated genes; upregulated genes were involved in Glu, Phe, Met, Gly Ile/Val/Leu, Asn/Asp and Ser degradation, whereas downregulated genes were responsible for the first assimilatory step of Met, Asp, Glu, Tyr, Ser and Ala. DEGs identified in the second contrast included genes involved in Glu and Gly degradation. Taking the DEGs indicated above as a proxy for the entire assimilatory process of the corresponding amino acids, we noticed a good overlap with available PhTAC125 physiological data[25].

DEGs analysis also allowed the identification of the major amino acid entry points into PhTAC125 metabolism. We counted, for example, 12 alternative possibilities steps to metabolize Asp in PhTAC125 (Fig. 2e), but the expression of only two of them (*purA* and *asnB*) appeared to be significantly modified during PhTAC125 growth. Similarly, six alternative steps can convert Glu to other cellular intermediates following its uptake. At T1, only two of these genes showed an altered expression level, suggesting that these may represent the most relevant players in Glu assimilation and usage. Nearly the same holds for Ser, with five distinct entry points and only two of them being differentially regulated.

Overall, we have identified possible key players, both in the switch among the set of metabolized amino acids, and in the entrance of amino acids into PhTAC125 entire metabolic network. However, nutrients switching requires an efficient genetic regulation to ensure that each catabolic pathway is active at the right moment, allowing a correct proteome allocation. For this reason, we analyzed the co-expression of genes belonging to the same metabolic pathway (Supplementary Note 3) and identified an overall dis-regulation of such genes (average Fisher's Z transformation average of Pearson correlation coefficient 0.49,

Fig. 3a, b, Supplementary Fig. 4 and Supplementary Table 4). Focusing on the known regulons including AA-genes (Supplementary Table 5), we noticed that nearly half of them (three out of seven) displayed a relatively low (0.47, ArgR) or almost absent (0.26 and 0.11, MetJ and TyrR1, respectively) correlation among the expression values of the corresponding genes (Table 1). Figure 3c–e summarizes the details of the correlation existing among each gene of each pathway. In the case of ArgR regulon, for example, the major contribution to the low intra-regulon correlation is due to *astA* and *astD* (PSHAa0195 and 0196, respectively), showing an almost opposite expression pattern compared with the other ArgR regulated genes, especially with PSHAa2287-91 (Fig. 3C). *astA* and *astD* are involved in the conversion of Arg to Glu, whereas ArgHA, B, C, F, G (encoded by PSHAa2287-91, respectively) are involved in the synthesis of Arg from Glu through the formation of citrulline and fumarate.

Concerning MetJ regulon, we noticed a group of genes (including PSHAa2222, PSHAa2223, PSHAa0287 and PSHAa2292) whose expression values are negatively correlated with those of genes PSHAa2274-76 and PSHAa1226 (Fig. 3d). This group of co-regulated genes include those involved in the conversion of homocysteine to Met (PSHAa2222 and PSHAa2223), an L-alanine-DL-glutamate epimerase (PSHAa0287) and a Methylthioribulose-1-phosphate dehydratase involved in the Met salvage pathway (PSHAa2292). Finally, as for TyrR1regulon, PSHAa2042-43, coding for 4a-hydroxytetrahydrobiopterin dehydratase and phenylalanine-4-hydroxylase are negatively correlated to the other genes in the same regulon (Fig. 3e). PSHAa2043 encodes *phhA* the gene responsible for the synthesis of Tyr from Phe, whereas PSHAa2042 (*phhB*) encodes a Pterin-4-alpha-carbinolamine dehydratase responsible for the conversion of 4a-hydroxytetrahydrobiopterin to dihydrobiopterin.

Upstream of most of the genes belonging to the three regulons considered, we were able to identify a conserved motif for each regulon (Fig. 3f–h), partially overlapping with their known TF binding site. Finally, a closer inspection to the metabolic steps encoded by the differentially regulated genes of these regulons revealed that they usually belong to different and symmetric regions of the same metabolic pathway. PSHAa0195 and PSHAa0196 respectively encodes for *astA* and *astB*, responsible for the first steps of the route leading to the formation of Glu from Arg. The other genes of the ArgR regulon are mostly involved in the production of Arg starting from Glu (Fig. 3i). Similarly, *phhA* (encoded by PSHAa2043) is involved in the formation of Tyr (from Phe), whereas all the other genes are responsible for the formation of Tyr from a set of different precursors (e.g., prephenate) (Fig. 3k). In the case of MetR regulon, among the genes that could be reliably assigned to the methionine metabolic pathway, one of the two group of co-regulated genes belong to the upper part of the pathway (upstream the main product methionine), whereas members the other one are in its close proximity (PSHAa2222) or involved in the methionine salvage pathway (PSHAa2292), the set of reactions responsible for the recycle of the thiomethyl group of S-adenosylmethionine from methylthioadenosine (Fig. 3j).

In a previous work[26], we have simulated the growth of PhTAC125 in a nutritionally complex environment (peptone) and derived the overall metabolic reprogramming occurring during growth in a rich undefined medium using constraint-based metabolic modelling (i.e., flux balance analysis, FBA) (Supplementary Note 4). We observed a good correlation between the measured changes in the expression of AA gene regulons and the predicted changes in metabolic fluxes of their encoded reactions using FBA (Pearson's product-moment correlation = 0.89, $p$ value = 0.019). See Supplementary Table 6 for further details.

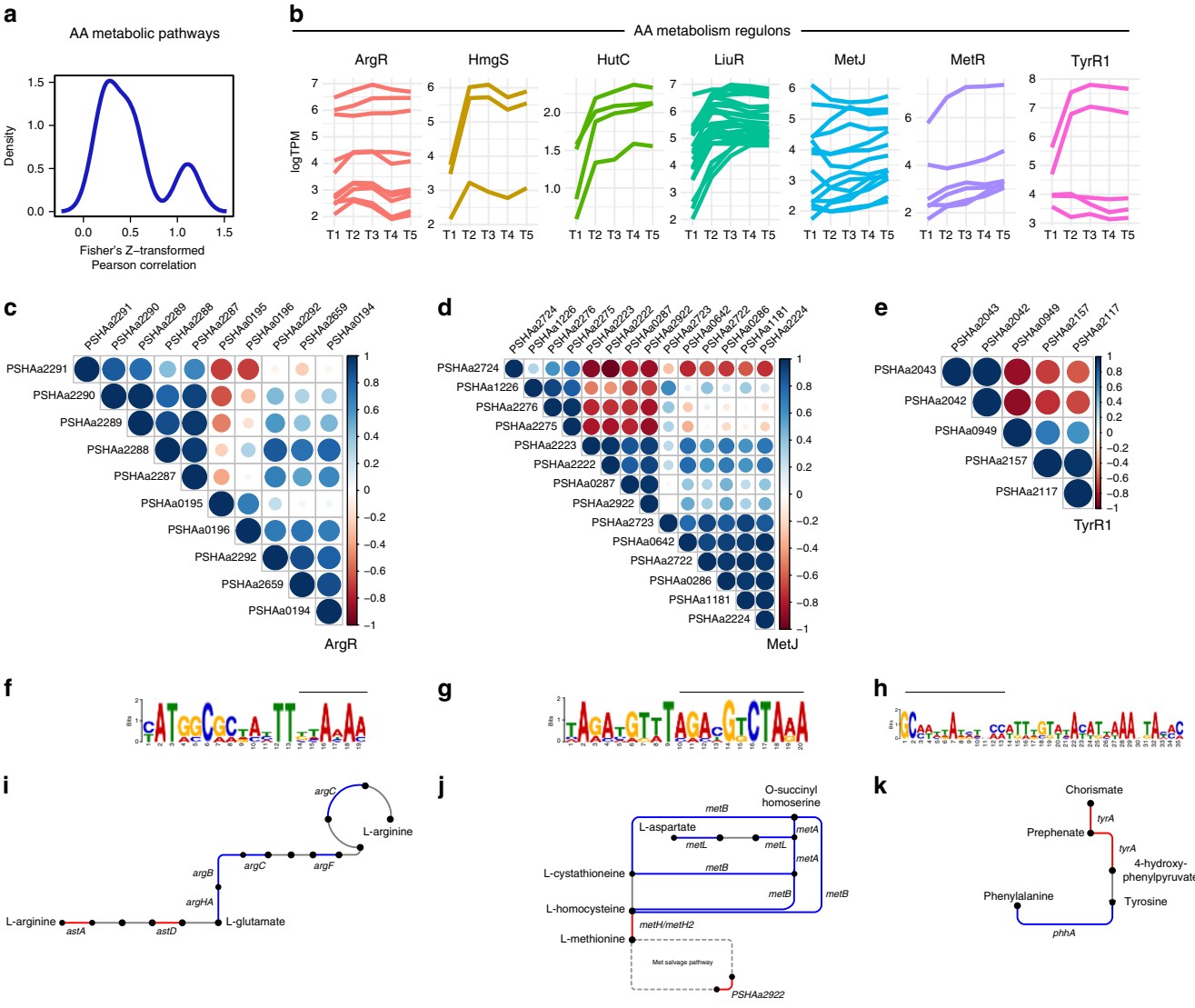

**Fig. 3 Deregulation of amino acid metabolic pathways. a** Cumulative density plot of Fisher's Z transformation average of Pearson correlation expressing the co-regulation of genes belonging to the same pathway. **b** Log$_2$-transformed TPM values of the genes belonging to the same regulon. **c–e** Graphic representation of the Pearson correlation matrix for the genes belonging to the regulons showing the lowest average correlation, Arginine metabolism/ biosynthesis, Methionine biosynthesis and Aromatic amino acids metabolism. Locus tags in bold indicate those of the regulator of the corresponding regulon. **f–h** Show the conserved upstream motif found for each gene of the considered regulon. The bar above each conserved motif indicates the overlap existing with the known TF binding site according to the RegPrecise database. Finally, in (**i–k**) the reactions encoded by the genes included in the same regulon are schematically represented. Red and blue lines schematically represent the correlation coefficient of Fig. 3c, d. Source data are provided as a Source Data file.

Taken together these results suggest that (i) the two growth lags observed (Fig. 1a) may be the same phenotypic representation of two different cellular states (i.e., assimilation strategies) and that (ii) a rather complex genetic regulation is at work to ensure a correct decision-making process in nutritionally dynamic environments. In the next sections these two aspects will be elucidated using controlled growth conditions and a combination of NMR experiments and theoretical modelling.

**Combination of simultaneous and sequential amino acids uptake.** Up to now, we have analyzed the behaviour of bacterial cells in a complex medium, using gene expression as a proxy for amino acids assimilation pathways. The medium used (peptone) is a complex mixture of nutrients whose exact composition is unknown.

Accordingly, it is not possible to conclude that the observed growth features (i.e., triauxic growth) are due to the exhaustion of certain preferred amino acids in the medium. For this reason, we assembled a medium including 19 amino acids, 0.2 mM each (named 19 AA medium, cysteine was not included in the list because of difficulties in its unambiguous quantitation during the experiments due to its spontaneous oxidation, as also reported in ref. [29]) and determined the kinetics of their usage during PhTAC125 growth by analyzing the growing media using $^1$H NMR (Supplementary Note 5 and Supplementary Fig. 5). Data obtained revealed that an important fraction of all the provided amino acids (16 out of 19) are consumed in the first 7 h of the growth. Afterwards, the remaining three amino acids (His, Met and Trp) are (slowly) metabolized (Fig. 4a). Clustering the amino acids assimilation profiles allowed a clearer visualization of the order in which amino acids are used by PhTAC125 during its

growth (Fig. 4b). This analysis divided the set of metabolized compounds into four main, non-overlapping clusters. Gln, Glu and Arg are the first amino acids to be consumed in the medium. Their concentration reaches (negligible) values close to 0.01 mM after 4.5 h of growth, remaining constant afterwards. The second set of amino acids is composed of Asn, Asp, Leu and Pro. Their degradation starts with a small delay with respect to the one of the first cluster, and they are completely removed from the

medium only between 6 and 6.5 h. The third cluster includes nine amino acids (Fig. 4b). Their consumption is rather slow in the first 3 h of growth; afterwards, it accelerates leading to negligible concentration of the corresponding amino acids at 7.5 h. The concentration of amino acids belonging to the fourth cluster remains overall constant for the first 6 h of growth. After that moment, corresponding to the point in which all the other amino acids are consumed, it starts decreasing. Importantly, this pattern of amino acids assimilation results in a triauxic growth curve (Fig. 4c). Indeed, (short) growth lag phases are observed after 4 and 6 h of growth, in correspondence with the major transition in amino acids assimilation pattern.

Overall, this behaviour highlights a balanced mix between simultaneous and sequential uptake of nutrients. Amino acids belonging to the same group (Fig. 4d) are simultaneously metabolized by the cells but the assimilation of different groups occurs with different dynamics and is responsible for growth lags in the curve. Finally, a typical diauxic nutrient shift is observed when all the main (preferred) sources are exhausted and the degradation of the other (previously ignored) compounds begins. Available growth phenotypes[24,30] (Supplementary Note 6) seem to suggest that the order in which nutrients are used by PhTaC125 during the growth depends both on the final biomass and on the specific growth rate achievable when grown with amino acids as sole carbon sources (Supplementary Fig. 6A, B). Also, the order of amino acids uptake, partially reflects their entry point into the TCA cycle (Supplementary Fig. 7).

**Table 1 Fisher's Z transformation average of Pearson correlation of the same-regulon genes.**

| Regulon | Process | Fisher's Z transformation of Pearson correlation coefficient |
|---------|---------|------------------------------------------------------------|
| HutC | His degradation | 2.55 |
| HmgS | Tyr degradation | 1.92 |
| MetR | Met biosynthesis | 1.22 |
| LiuR | Branched-chain amino acid degradation | 0.97 |
| ArgR | Arg biosynthesis/ degradation | 0.47 |
| MetJ | Met metabolism, Met degradation | 0.26 |
| TyrR1 | Aromatic amino acids metabolism | 0.11 |

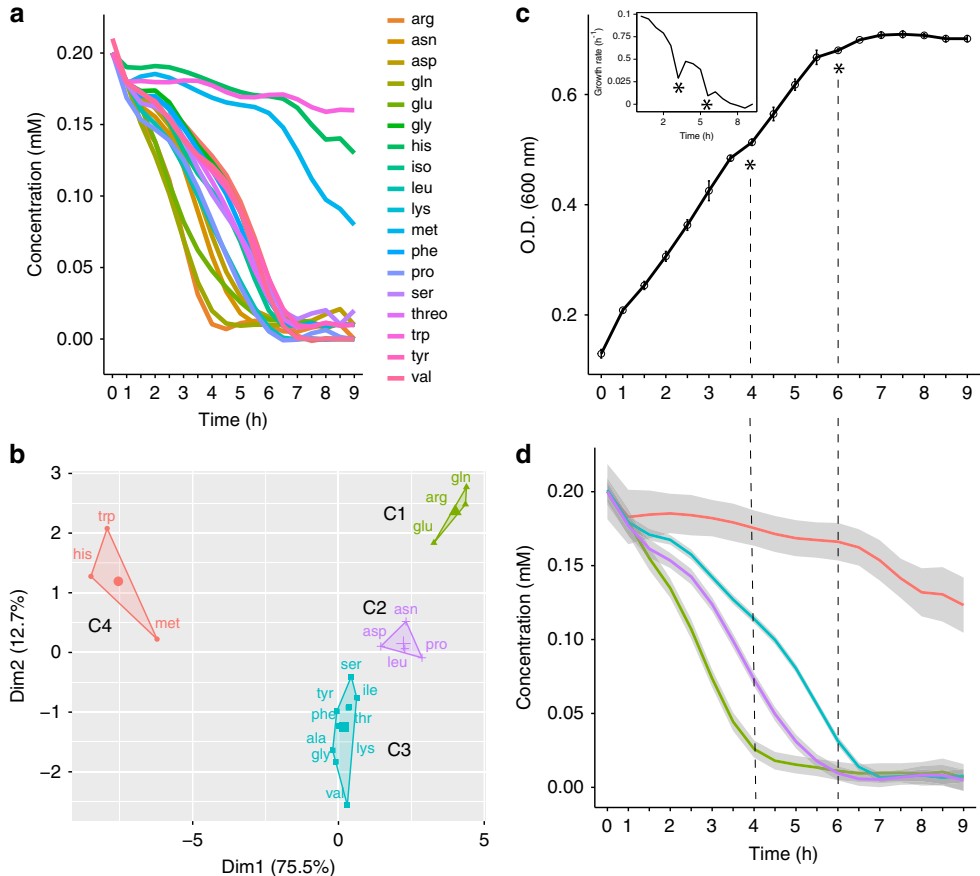

**Fig. 4 Amino acids assimilation profiles. a** Degradation dynamics for each of the 19 amino acids included in the defined AA medium. **b** Clustering of time-resolved concentration values for the 19 amino acids analyzed. Error bars represent SD of two different cell cultures in two independent experiments. **c** Growth curve of the 19 amino acids experiment. Asterisks indicate growth lags. **d** Degradation dynamics for each of the four identified clusters of amino acids included in the defined AA medium. Colour codes as in **b**. Grey shaded area includes the 95% confidence of the linear regression (coloured) line over the concentrations of the amino acids belonging to the same group. Source data are provided as a Source Data file.

Despite the trend seems to be quite clear, a certain variability was observed in the results shown in Supplementary Fig. 6A, B. This is the case, for example, of Leu in cluster C2 that is preferred to Ala (C3) despite the growth rate of PhTAC125 growth on Leu is about twofold lower than that on Ala. This might be accounted for by promiscuous uptake of nutrients. Indeed, the broad spectrum of some amino acids transporters is well described in bacteria[31,32]. Thus, some amino acids might be taken up from the medium not as the result of an active cellular control over the most efficient carbon sources, but as the result of the broad-spectrum activity of a membrane transporter.

An additional explanation to the composition of the clusters might involve the overlap among the catabolic pathways of the substrate that are co-consumed (i.e., belonging to the same cluster, Supplementary Note 7). Indeed, we noticed that C1 amino acids (Arg, Glu and Gln) share part of their catabolic pathways and are all converted to alpha-ketoglutarate before entering the TCA cycle. Two (out of four, i.e., Asp and Asn) amino acids of cluster C2 are converted to oxaloacetate before entering the TCA cycle. Most of the C3 amino acids (six out of nine, i.e., Ala, Ser, Gly, Thr, Phe and Tyr) are converted to pyruvate, thus not being directly catabolized into one of the TCA cycle intermediates. Other three C3 amino acids (Thr, Ile and Val) can be catabolized to form pyruvate and a TCA intermediate (i.e., Succinyl-CoA). Finally, C4 includes those amino acids that are negligibly used during PhTAC125 growth, so no degradation pathways overlap is required to explain their inclusion in the same group (Supplementary Fig. 7).

In order to check whether the sequential or co-consumption of substrates also depends on their own concentrations, we performed a set of additional experiments aimed at evaluating the effect of higher concentrations of nutrients on the pattern of nutrients assimilation (Supplementary Note 8). We both tested the effects of (i) providing all the 19 amino acids at concentrations five and ten times higher (1 and 2 mM, respectively) than those used in the 19 AA experiment described above and (ii) increasing the concentration of late-metabolized amino acids (His, Met, Trp, 2 mM) on the timing of the last metabolic switch observed during PhTAC125 growth. The results of these analyses are reported in (Supplementary Figs. 8 and 9) and revealed that metabolic phenotype identified in the original 19 AA experiment is poorly affected by the concentration of the available carbon sources. Also, we compared the phenotype observed in PhTAC125 with the one of a model organism (E. coli, Supplementary Note 9), when grown in the same nutritional environment (19 AA medium, 0.2 mM each). A poor overlap was observed between the specific response to nutrients switching of the two microorganisms (Supplementary Figs. 10 and 11).

**The different fate of the catabolized amino acids**. Next, we investigated the different fate of amino acids belonging to the same cluster once entered inside PhTAC125 cells. To this aim, we used uniformly labelled $^{13}$C amino acids (Glu, Asp, Ala and Met, belonging to C1, C2, C3 and C4, respectively) and followed the path of their labelled carbon atoms inside the cells by NMR. More in detail, for each labelled amino acid used, we prepared four different parallel cultures, each containing 18 amino acids plus the $^{13}$C labelled one as the only carbon source for PhTAC125. These four cultures were run in parallel and each of them was sacrificed at a different time point. Specifically, we analyzed four time points, i.e., early and late exponential growth (3 and 6 h) and early and late stationary phase (8 h and 30 min, 24 h). For each time point we analyzed both growing media and the cell lysates by acquiring mono-dimensional $^1$H NMR spectra and bidimensional $^1$H-$^{13}$C HSQC NMR spectra. A scheme of the structure of

this experiment is reported in Supplementary Fig. 12. This experiment allowed us to study the metabolic fluxes in a time-dependent fashion and provided hints on the fate of the metabolized amino acids inside the cell (Fig. 5).

First, by using $^1$H NMR spectra of growing media, we confirmed the same growth dynamics for all the replicates of the experiment and the same overall growth features observed in the unlabelled 19 AA experiment (compare Fig. 5a, c). Similarly, amino acids were consumed in the same order and with the same overall rates previously observed (compare Figs. 5b and 4a).

A principal component analysis (PCA) performed on the $^1$H spectra of cell lysates clustered the samples according to their sampling time, thus confirming the consistency and the high reproducibility among the different replicates (Supplementary Fig. 13), besides the occurrence of significantly different metabolic profiles of PhTAC125 cells along the four sampled time points.

$^1$H-$^{13}$C NMR spectra of cell lysates were used to have an overview of the fate of the metabolized amino acids inside the cells. Each of the four selected amino acids showed a specific path of labelled carbon atoms inside the cells, as revealed by the PCA reported in Fig. 4c ($^1$H-$^{13}$C HSQC NMR spectra). While the spectra of Asp and Glu almost overlap, the spectra of Ala and Met are very well separated. This trend probably reflects the fact that both Glu and Asp are used to directly feed the TCA cycle, whereas Met and Ala are used to feed different pathways inside the cell (see below).

In particular, we were able to show that Ala is mostly used to feed many important pathways inside the cell, namely glycolysis/gluconeogenesis, nucleotide precursors metabolism and (partly) TCA cycle. At T3 we found $^{13}$C coming from Ala degradation in most of the key compounds that are the input/output of the aforementioned pathways, i.e., pyruvate, phosphoenolpyruvate (PEP), fatty acids and AXP/GXP (Fig. 5d, Supplementary Figs. 14 and 15). Oxaloacetate is the only TCA cycle intermediate that displays Ala labelling at T3 (Fig. 5d and Supplementary Fig. 16). At T3, we found Asp-derived $^{13}$C labelling on all the TCA cycle intermediates identified and on purine biosynthesis intermediates, AXP and GXP (Fig. 5d, Supplementary Figs. 15 and 16). At later growth stages Asp labelling appeared also on PEP and NAD precursors (Fig. 5d, Supplementary Figs. 14 and 17). Glu-derived $^{13}$C labelling appears in all the TCA intermediates analyzed (with the exception of oxaloacetate) starting from T3 (Fig. 5d and Supplementary Fig. 16), suggesting that Glu is readily redirected towards the TCA upon its uptake. Interestingly, Glu seems also to be used as a substrate for NAD (and precursors) biosynthesis from the early stage of growth (Fig. 5D and Supplementary Fig. 17). Finally, this experiment confirmed that Met is incorporated into the PhTAC125 metabolic network at a later stage of its growth, in that no identified compounds was labelled with $^{13}$C of Met at T3. Met labelled $^{13}$C appear to be included into homocysteine starting from T6. After 24 h, we found Met labelled $^{13}$C on a compound whose NMR pattern corresponds to that of trimethylamine (Fig. 5d and Supplementary Fig. 18). Interestingly, in a previous work[33], we had characterized PhTAC125 as a methylamine producer and also showed that adding Met to the growth medium was pivotal to allow PhTAC125 to produce this compound (and inhibit the growth of human opportunistic pathogens). The metabolic flux analysis performed here suggests a link between the production of methylamine from the degradation of Met, possibly through the formation of trimethylamine.

Overall, using $^{13}$C-labelled precursors provided evidence that the different amino acids used by PhTAC125 during its growth have different and complementary roles within the cell. Ala is readily converted into pyruvate and this is used to feed both gluconeogenic pathways and (possibly through the formation of

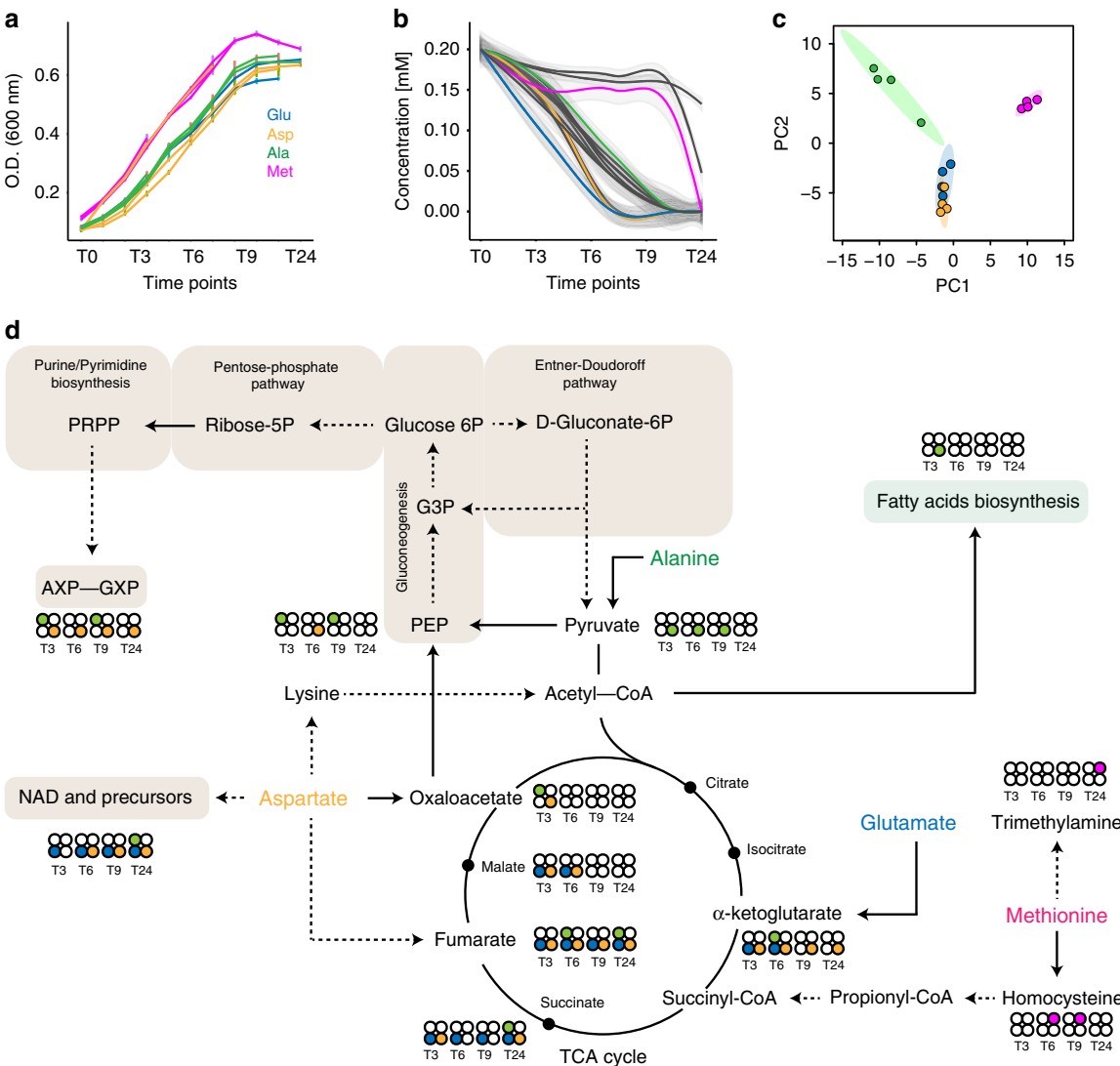

**Fig. 5 The fate of catabolized amino acids. a** PhTAC125 growth curves across all replicates in the 19 AA medium with one of them being $^{13}$C labelled. Each colour indicates which was the $^{13}$C-labelled amino acids for each replicate. Error bars represent SD of four different cell cultures. **b** Extracellular concentration of all the amino acids in time. Colour codes as in **a**. **c** PCA on the $^{13}$C spectra. Colour codes as in **a**. Grey shaded area includes the 95% confidence of the linear regression (coloured) line over the concentrations of the amino acids belonging to the same group. **d** Central PhTAC125 metabolic network, highlighting the possible entry points of the labelled amino acids. Circles close to compounds indicate the $^{13}$C signal recovered at the different time points (T3–T24) and coming from different amino acids (colour codes as in **a**) corresponding. Source data are provided as a Source Data file.

acetyl-CoA) fatty acids synthesis. Only at later growth stages, TCA intermediates started to display Ala-derived labelling. Conversely, Glu is immediately used to feed the TCA cycle as all its intermediates but one (oxaloacetate) carry Glu labelling at T3. No other compounds in the upper part of Fig. 5d displayed Glu-derived $^{13}$C atoms, thus suggesting a clear separation between Ala and Glu assimilation pathways. The primary assimilation pathway of Asp seems to be through the TCA cycle, although part of its carbon skeleton is diverted to PEP formation (and possibly to gluconeogenesis) at T6. Quite interestingly, $^{13}$C atoms derived from Asp degradation are observed on purine metabolism product AXP and GXP from the first sampled time point to the last, despite the gluconeogenic metabolism precursor PEP does not display the same signal. We thus argue that the origin of Asp-derived labelling on AXP and GXP might be due to TCA intermediates playing a role in purine metabolism, e.g., 5-phosphoribosylamine, Finally, we confirmed the late entrance of Met into PhTAC125 metabolic network, being initially converted to homocysteine and then also rerouted towards the production

of trimethylamine (at T24), with a possible but still undisclosed pathway involving the formation of betaine or carnitine[34–36].

To further characterize the metabolic response of PhTAC125 to nutrients consumption, the whole intracellular metabolome was evaluated through untargeted $^1$H NMR across four time points, i.e., early and late exponential growth (3 and 6 h) and early and late stationary phase (8 h and 30 min, 24 h). In the spectra, the signals of 26 metabolites could be unambiguously assigned and quantified (Supplementary Fig. 19 and 20). These metabolites were representative of six major metabolic modules, i.e., amino acids metabolism, purine and pyrimidine metabolism, sugars, amino sugars metabolism, nucleotide precursors and TCA. The trend of metabolites' concentration in time revealed that the relative concentration of purine and pyrimidines intermediates (namely uridine and inosine) decreases in time. A similar trend is observed for ribose and for the detected TCA intermediate (fumarate). This scenario is compatible with the decrease in growth rate and overall biomass production of PhTAC125 along the growth curve (Fig. 4c) and with a consequent decrease of DNA

synthesis activity (purine and pyrimidine precursors) and energy demand (TCA cycle). Conversely, sugars (e.g., glucose) and amino sugars (e.g., UDP-NAG) increase their cellular concentration in the latest growth stages, reaching values that are up to three times the initial ones. This latter finding can be interpreted in the light of two considerations: first, we have already shown that carbohydrates metabolic genes increase their expression at the final stages of PhTAC125 growth on peptones, suggesting the activation of sugar metabolism-related pathways at later growth stages. Second, the increase of sugar/amino sugars intracellular concentrations might have a role in PhTAC125 cell aggregation when nutrients concentration starts to deplete given that: (i) such metabolites are known to be involved in cell–cell contacts[37], (ii) PhTAC125 was shown to produce a biofilm that incorporates amino sugars and (iii) this is supposed to be a strategy to survive in poor nutrient conditions[37]. Our findings are in line with a scenario in which the presence of a lower availability of nutrients induces a greater production of biofilm since the biofilm matrix can improve the capture of nutrients[38].

**Modelling simultaneous and sequential amino acids uptake**. At least two different explanations may account for the mixed sequential/diauxic nutrient uptake. Either this phenotype is "simply" determined by different uptake kinetics of the different compounds or the assimilation pattern is actively regulated by the cells. To discern between these two scenarios, we implemented two mathematical models accounting for cell growth and nutrients uptake during the 19 AA experiment. To reduce the complexity of the problem, the 19 amino acids were lumped into the four corresponding clusters shown in Fig. 4b, d. In this way we modelled the growth of PhTAC125 in a hypothetical growth medium embedding four different groups of carbon sources ideally representing the 19 AA medium. The first model is based on the Michaelis–Menten–Monod kinetics (MMM model) and is formulated as follows:

$$P + S_i \xrightarrow{r} 2P, \tag{1}$$

where $P$ represents bacterial cells, $S_i$ (with $i = 1, 2, 3, 4$) represents each of the four groups of pooled C sources and $r$ the rate at which the reaction occurs. Specifically, $r$ was modelled according to a canonical (Michaelis–Menten-derived) Monod kinetics with:

$$r = \frac{\beta_i \cdot \phi S_i}{k_i + \phi S_i}, \tag{2}$$

where $\beta_i$, $\phi S_i$ and $k_i$, represent the maximum rate constant for cell production, the concentration and the Michaelis–Menten constant for the $i$th group of amino acids, respectively. According to these formulas, the state variables model can be written as:

$$\frac{d\phi P}{dt} = \sum_{i=1}^{4} \frac{\beta_i \cdot \phi S_i^2}{k_i + \phi S_i} \cdot \phi P - d \cdot \phi P, \tag{3}$$

$$\frac{d\phi S_i}{dt} = -\frac{\beta_i \cdot \phi S_i^2}{k_i + \phi S_i} \cdot \phi P, \tag{4}$$

where $S_i$ (with $i = 1, 2, 3, 4$) represents each of the four lumped substrates and $d$ the bacterial cells death rate.

The second model implemented here accounts for the effect of the regulatory processes of catabolite inhibition and activation that can be observed during microbial growth on multiple substrates. The model is a modified version of the cybernetic model proposed by ref. [39], overall resembling the one proposed in ref. [40].

The cybernetic modelling framework takes into account the (yet) unknown regulatory processes regulating the microorganisms' uptake kinetics. It assumes that microorganisms have evolved under the selective pressure to become optimal with respect to certain cellular objectives (in our case, maximization of

biomass production) and achieve this task by actively modulating the induction/repression and activation/inhibition of the key enzymes of substrates available in their external environment. Cybernetic variables (see below) are introduced in the model to account for the induction/repression and activation/inhibition of the key, bottleneck enzymes regulating cell growth, substrate consumption, and key enzyme production[41,42].

According to this model, the assimilation of substrate $S_i$ by cells $P$ (Eq. (1)) is assumed to be catalyzed by the set of enzymes $E_i$ (with $i = 1, 2, 3, 4$). The assumption here is that enzymes responsible for the assimilatory pathway of each pool of nutrients are induced by the presence of $S_i$ (and repressed by the presence of the other nutrients). This alternative model can be written as:

$$P \xrightarrow{r_i v_i} 2P, \tag{5}$$

$$\emptyset \xrightarrow{r_{E_i} u_i} E_i, \tag{6}$$

where $E_i$ represents the key assimilatory enzyme. The rate equations for biomass production (Eq. (5)) and for enzyme synthesis (Eq. (6)) can be written as a modified form of Monod's equation and are respectively expressed as follows:

$$r_i = V_{max,i} \cdot \phi E_i \cdot \frac{\phi S_i}{K_{S_i} + \phi S_i}, \tag{7}$$

$$r_{E_i} = V_i \cdot \frac{\phi S_i}{K_{E_i} + \phi S_i}, \tag{8}$$

where $\phi E_i$ represents enzyme concentration, $V_{E_i}$ is the maximum rate constant for enzyme's biosynthesis and $V_{max,i}$ is the is the maximum rate constant for bacterial production $P$ on the $i$th substrate. $K_{S_i}$ and $K_{E_i}$ are the Michaelis–Menten constant for the $i$th substrate and the synthesis of the $i$th enzyme, respectively.

The inhibition/activation effect due to the concentration of the different substrates is accounted for by two (control) variables, $u_i$ and $v_i$, representing the fractional allocation of resource for the synthesis of $E_i$ and the mechanism of controlling enzymes $E_i$ activity, respectively. $u_i$ is expressed as:

$$u_i = \frac{r_i}{\sum_{j=1}^{4} r_j}, \tag{9}$$

with $0 \leq u_i \leq 1$ and $\sum_{j=1}^{4} u_i = 1$. The other control parameter, $v_i$ is expressed as:

$$v_i = \frac{r_i}{max\{r_1, r_2, r_3, r_4\}}, \tag{10}$$

where the denominator accounts for the observation that priority is given to the consumption of the substrate(s) that guarantee the highest growth rate[43]. The model further considers constitutive enzyme production rate ($\beta_i$), the effect of dilution of the specific enzyme level due to cell growth ($\alpha$), constant protein decay in the cells and bacterial death rate ($d$) and can be written as follows:

$$\frac{d\phi S_i}{dt} = -r_i \cdot v_i \cdot \phi P, \tag{11}$$

$$\frac{d\phi P}{dt} = \sum_{j=1}^{4} v_j \cdot r_j \cdot \phi P - d \cdot \phi P, \tag{12}$$

$$\frac{d\phi E_i}{dt} = u_i \cdot r_{E_i} - \sum_{j=1}^{4} v_j \cdot r_j \cdot \phi P - \alpha \cdot \phi E_i + \beta_i. \tag{13}$$

The model parameters were determined by fitting the experimental data (shown in Fig. 4a–d) with model simulations and their values are reported in Supplementary Tables 7 and 8. As

shown in Fig. 6, the cybernetic model accurately reproduces the dynamics of all the species considered. This is even clearer in the case of nutrient concentration dynamics where the model implementing the MMM model, is not capable of producing a satisfactory approximation of the real data. Indeed, $R^2$ calculation indicates that the cybernetic modelling framework is performing better on four out of five of the species included in the model (Table 2). Likewise, when using the AIC metric, the comparison between cybernetic model and MMM was shown to follow the same trend.

These results suggest that, in the conditions tested, the uptake of nutrients is tightly regulated, leading to the simultaneous presence of diauxic and co-utilization strategies within the same growth curve. Hints on the main catabolic players involved in such assimilation patterns were obtained combining transcriptomic data from the complex-medium experiment and RT-PCR on specific targets (see Supplementary Note 10, Supplementary Fig. 21 and Supplementary Table 9).

## Discussion

Our knowledge on the possible bacterial strategies for nutrients assimilation when multiple sources are available is biased by the fact that it has been mainly studied in a few model organisms, providing them with a reduced number of possible inputs (compared with those available in their source environment). Here, we have characterized a non-model response to nutrients switching and studied the process of bacterial nutrients uptake in experimental conditions that more closely resemble a natural setting, in terms of the availability of many different substrates simultaneously. Using a marine heterotrophic bacterium (*P. haloplanktis* TAC125) as a case study strain, we have shown that its response during growth lags do not resemble the one currently known for *E. coli*. Only 10% of the *E. coli* metabolic regulators and two (out of eight) main generic controllers (*rpoS* and *rpoD*) displayed an altered expression level in our experiments. Also, we showed that when the two microorganisms were independently cultivated in the same defined medium embedding 19 different amino acids, differences arose in the choice of the amino acids to utilize, in the timing of uptake and in the presence/absence of overall growth lags.

The poor overlap between PhTAC1215 and *E. coli* transcriptional response suggests that, in marine bacteria, the response to nutritional switches and/or multi-auxic growth patterns may involve still untapped genetic circuits. As a matter of fact, PhTAC125 is known to lack the CCR system[44], which is currently referred to as the main driver in metabolic switches and diauxic phenotypes. Using time-resolved transcriptomics we have shown that growth lags in a nutritionally complex environment are probably due to the exhaustion of specific carbon sources and that such event has a deep impact also on other important gene categories including, for example, motility-related genes. In the second part of the curve, i.e., when nutrient concentration decreases, cell motility genes increase their expression, probably reflecting the need to explore the surrounding environment for other potential sources. This is in line with the observation that

many bacteria become motile when nutrients are scarce[45]. Moreover, among the genes peaking their expression in correspondence of the first growth lag, those involved in post-translational modification, protein turnover and chaperon are

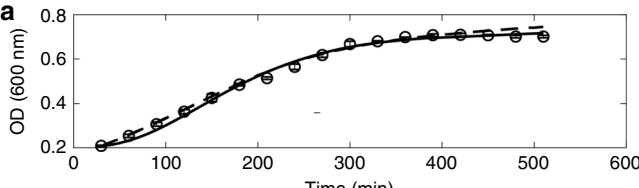

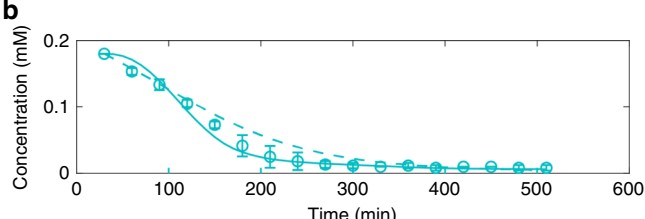

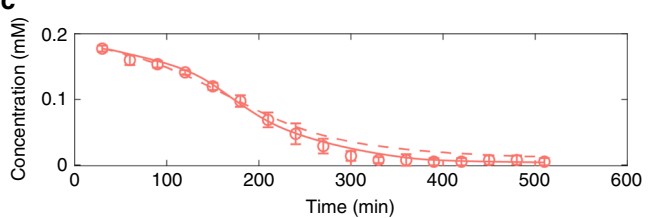

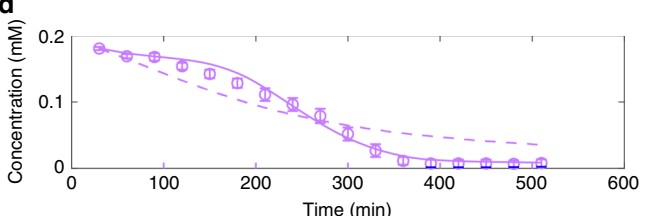

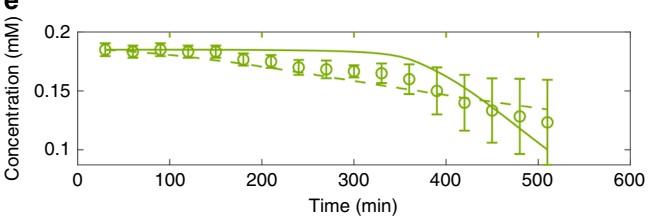

**Fig. 6 Modelling the multi-auxic growth.** Simulation outcomes for the two models implemented here for biomass (**a**) and lumped nutrients (**b–e**). Dashed lines represent the prediction for the MMM model. Solid lines represent the prediction for the cybernetic model. Open circles represent experimental data (same data shown in Fig. 4d). Error bars represent SD of two different cell cultures in two independent experiments.

**Table 2 $R^2$ and AIC calculation for the different simulations using the two different models.**

| Model | Cluster A | Cluster B | Cluster C | Cluster D | Biomass |
|---|---|---|---|---|---|
| Pearson—MMM model | 0.9844 | 0.9922 | 0.9584 | 0.9584 | 0.9959 |
| Pearson—Cybernetic | 0.9920 | 0.9973 | 0.9953 | 0.9193 | 0.9967 |
| AIC—MMM model | −408.53 | −418.46 | −338.17 | −476.82 | −378.46 |
| AIC—Cybernetic | −439.48 | −475.38 | −438.75 | −395.97 | −372.37 |

significantly over-represented. Generally speaking, proteins belonging to this functional category can be easily associated to the stress encountered by an exponentially growing batch culture that exhaust (part of) the readily available and preferred nutrients in the growth medium. At this moment cells undergo a regulated transition into stasis by activating a stereotypic stress response. Post-translational modifications have been shown to play a role in the starvation-induced growth arrest, for example in *S. coelicolor*[28,46]. It is to be noticed that these proteins do not increase their expression in the later stages of the growth and, in particular, in the second growth arrest experienced by PhTAC125. This, together with the observation that both the overall number of DEGs and of those involved in many other functional processes (amino acids degradation, TFs and TCRS) is much higher in correspondence of the first time contrast, suggested that the same phenotype (the two growth lags) were mirrored by profoundly different cellular reprogramming patterns. This prompted us to investigate more in depth the regulation of amino acid assimilation pathways. This analysis highlighted the tight regulation required to efficiently exploit complex and amino acid enriched, nutritional conditions. A paradigmatic example of this capability is the (dis)regulation of several genes belonging to the same regulon (ArgR, TyrR1 and MetR), responsible for the activity of distinct and conflicting functional metabolic modules inside the same metabolic pathway. In principle, all the genes belonging to the same regulon are under the control the same TF. It is known that some TFs may function as either activators or repressors, often according to the positioning of the TF binding site in the target promoter, although this feature had not been described, to date, for amino acid assimilation pathways[47]. Apparently, such a mechanism is at work in some of the amino acid catabolic pathways of PhTAC125, probably ensuring an efficient and correct exploitation of the amino acid mixture available in the surrounding environment.

Despite being closer to the environmental natural setting, growth in complex medium does not allow a precise understanding of the usage of all the available carbon sources. For this reason, we have assembled a defined but nutritionally rich medium containing 19 amino acids as the sole carbon sources for the bacterial cells. Tracking their concentration in time, we showed that the two main feeding strategies commonly thought to be exclusive to each other (i.e., sequential and simultaneous uptake of nutrients) can coexist in the same growth curve. Clustering metabolized amino acids into four major groups revealed that amino acids belonging to the same group are co-utilized, whereas the switch among the different clusters is tightly modulated. A "canonical" diauxic shift is finally observed at the end of the growth, when the consumption of a set of previously untapped nutrients begins. Thus, similar to the complex-medium growth experiment (Fig. 1), the two growth lags apparently underlay different cellular states. Indeed, despite the exhaustion of nutrients is common to both growth lags, in one case (the first lag) cells are already metabolizing alternative compounds when the exhaustion of the preferred sources occurs.

The order in which nutrients are utilized can be partially explained by the biomass yield and growth rates obtained when each single amino acid is provided as single carbon sources. Those amino acids allowing the highest growth rate and biomass production are those that are consumed first in the 19 AA experiment.

Using [13]C-labelled amino acids and time-resolved NMR spectra, we were able to follow the degradation pathways of four selected amino acids (Glu, Ala, Asp and Met) and derive their fate inside the cell. We showed that Ala is readily converted into pyruvate and then probably used to fuel sugar metabolism (i.e.,

gluconeogenesis and pentose phosphate pathway), leading to the production of ribonucleotides. Asp and Glu are instead promptly used to fuel the TCA cycle, with two notable exceptions, i.e., the entrance of Glu-derived carbons into the biosynthesis of NAD precursors and the conversion of (part of) the initial amount of Asp into PEP and its usage for ribonucleotide biosynthesis. The fate of Met inside the metabolic network remained hard to decipher, and the actual contribution to the growth of PhTAc125 will require further investigation. Further, results coming from the untargeted evaluation of the overall intracellular metabolome is in line with an increased importance of sugar metabolism/ intermediates upon exhaustion of the available amino acids in the medium, probably reflected in the characterized production of biofilm in poor nutrients conditions.

Finally, we have shown that the dynamics of nutrients degradation can be explained using a theoretical model that accounts for gene regulation and, in general, for the proper resource allocation for the synthesis of the main assimilatory pathways. This modelling framework can accurately interpret the pattern of nutrients degradation in a nutritionally rich environment.

In conclusion, we would like to stress the importance of cultivating and studying microorganisms in nutritional conditions that more closely resemble the ones most found in nature, for example for what concerns the contemporary availability of many distinct possible carbon sources as done here. By doing so and using a combination of computational and experimental (transcriptomics and NMR-based metabolomics) approaches, we have shown that, despite diauxie and co-utilization strategies have been usually thought as conflicting phenotypes, they can coexist in the same growth curve and give rise to a diversified ensemble of feeding strategies.

The use of different sources depending on the phase of cell growth and, most of all, a distinct metabolic fate inside the cell for each of the metabolized compounds, is a common feature of intracellular bacteria (e.g., *Legionella pneumophila*, *Listeria monocytogenes* or *Coxiella burnetii*[17,19,48–50]) suggesting that plastic strategies for carbon assimilation might be evolved in response to nutritionally poor and highly variable conditions.

It will be interesting to investigate which are the molecular mechanisms allowing the implementation of this mixed and apparently unconventional feeding strategy and, in particular, the fine-tuned regulatory circuits that are probably responsible for the efficient switching among all the available carbon sources. Future efforts will be also devoted to understanding the effect of fluctuations (in the number of cells and/or in nutrients concentration) and of possible population heterogenicity[10,11] on the resulting growth dynamics of heterotrophic marine bacteria.

## Methods

**Bacterial strain, media and growth condition**. *P. haloplanktis* TAC125[51] cells were routinely grown in Marine Agar (MA) or Broth (MB) (Condalab, Spain) under aerobic condition at 21 °C. The stock suspension of the strain was stored in 20% [v/v] glycerol solution at −80 °C. For growth curves experiment, Schatz salts[52] (1 g/l KH$_2$PO4, 1 g/l NH$_4$NO3, 10 g/l NaCl, 0.2 g/l MgSO$_4$ × 7H$_2$O, 0.01 g/l FeSO4 × 7H$_2$O, 0.01 g/l CaCl$_2$ × 2H$_2$O) were supplemented with 5 g/l Peptone N-Z-Soy BL 7 (Sigma-Aldrich S.r.l) (complex medium) or with 19 amino acids (19 AA medium) each one at a final concentration of 0.2 mM (cysteine was not included due to its rapid oxidation to cystin[29]). To confirm the order of AA we used Schatz salts supplemented with (i) 19 AA at a final concentration of 1 mM each, (ii) 19 AA at a final concentration of 2 mM each and (iii) 16 AA at a final concentration of 0.2 mM and histidine, methionine and tryptophan at a final concentration of 2 mM. The experiments with [13]C AA were performed using Schatz salts supplemented with 18 standard AA and one of the four marked amino acid (L-Alanine-[13]C$_3$, L-Aspartic acid-[13]C$_4$, L-Glutamic acid-[13]C$_5$ and L-Methionine-[13]C$_5$) all at a final concentration of 0.2 mM. In all cases the pH was adjusted to 7.0. All the amino acids were purchased from Sigma-Aldrich S.r.l. All the experiments were performed under aerobic condition at 21 °C.

*E. coli* Dh5α (laboratory stock suspension stored in 20% [v/v] glycerol solution at −80 °C) were grown in Luria Bertani (LB)[53], agar or broth, and in M9 media[53]

supplemented with 19 AA at a final concentration of 0.2 mM (pH 7.0) (19 AA M9 medium). The experiments were performed under aerobic condition at 37 °C.

**Growth curve experiments.** The growth curve experiments with PhTAC125 were performed after two pre-cultures, as in Wilmes et al.[25] with some adaptations. For the complex medium growth curves, a first preculture was grown for 20 h, in 20 ml MB medium in a 100 ml flask. Then this preculture was diluted 1:1000 in a final volume of 100 ml of the complex medium in a 1 l flask. After 20 h of growth, the optical density ($OD_{600}$) of this second preculture was measured to be used to inoculate the final flask (1 l) in a final volume of 150 ml of Schatz salts and Peptone with a starting $OD_{600}$ ~ 0.1. For the growth curves in the 19 AA medium, for all the concentrations used, the first preculture in MB was diluted 1:100 in a final volume of 100 ml of the 19 AA medium (for each experiment the AA concentration used for the final growth curves was used) in a 1 l flask. After 22 h of growth, the second preculture was washed, resuspend and used to inoculate the final flask (1l) in a final volume of 200 ml of the 19 AA medium with a starting $OD_{600}$ ~ 0.1. In all experiments, the pre-cultures and the final growth cultures were incubated at 21 °C with shaking. Each experiment was performed in duplicate. Cell growth was monitored measuring the $OD_{600}$ every hour in the experiments in complex medium, and every half an hour or an hour, depending on the concentration of AA used, in the experiments with the 19 AA medium. Three different measures were performed at each time point for each biological replicate.

The growth curve experiments with *E. coli* were performed after a first preculture of 20 h in 20 ml of LB broth in a 100 ml flask and a second preculture in 100 ml of 19 AA M9 medium.

After 20 h of growth, the optical density ($OD_{600}$) was measured to be used to inoculate the final flask (1 l) in a final volume of 200 ml of 19 AA M9 medium. The cells were grown at 37 °C with shaking in duplicate and the growth was monitored measuring the $OD_{600}$ every hour.

**Sampling.** Two biological replicates of the growth curves performed in complex medium were used for RNA-seq experiment. Every hour, in correspondence of the $OD_{600}$ measurements, two replicates of 500 µl each for each curve, were treated with the RNA protect bacteria reagent (Qiagen S.r.l.) and conserved at −80 °C.

During the experiment in the 0.2 mM 19 AA medium, at each time point, two replicates of 500 µl each for each curve, were treated with the RNA protect bacteria reagent like above. During all the experiment in the 19 AA medium, regardless of the concentration of AA used, and in the growth curve with *E.coli*, two replicates of 1 ml each for each curve were filtered at each time point (Filtropur 0.2 µm, SARSTED AG & Co. KG) to remove bacterial cells and conserved at −20 °C for NMR metabolomic.

**Growth curves with $^{13}$C amino acids.** Four different experiments were performed using one uniformly $^{13}$C-labelled amino acid each time ($^{13}$C-Glu, $^{13}$C-Ala, $^{13}$C-Asp and $^{13}$C-Met). For each experiment, two pre-cultures as described above were used, while the final growth experiments were performed in quadruplicate. Cell growth were monitored measuring the $OD_{600}$ every hour. At four time points, early and late exponential growth (3 and 6 h) and early and late stationary phase (8 h and 30 min, 24 h), one of the four replicates was analyzed. Overall 1 ml of the medium were filtered (Filtropur 0.2 µm, SARSTED AG & Co. KG) to remove bacterial cells and conserved at −20 °C. The remaining culture (199 ml), was pelleted by centrifuging for 10 min at 11,000 rpm at 4 °C and resuspended in 500 µl of PBS[54]. Then, cells were sonicated for 20 min, with cycle of 1 s of activity and 9 s of rest (292.5 W, 13 mm tip), with contemporary cooling on ice. After lysis, the samples were centrifugated for 25 min at 4 °C, at 8000 g, as described in ref. [54].

**RNA extraction and sequencing.** For RNA-seq, a preliminary sequencing (data not shown) was performed on an Illumina Hiseq 50 platform (Genomix4Life S.r.l., Italy). Total RNA was extracted with a RNeasy Tissue Mini Kit (Qiagen S.r.l.) following manufacturer's instructions. For improving the lysis step proteinase k and lysozyme were added to the lysis solution and the samples were homogenized using Tissue lyser II (Qiagen S.r.l.). The concentration and purity of RNA were analyzed using a NanoDrop ND-1000 (Thermo Fisher Scientific) and a Bioanalyzer (Agilent Technologies, Inc.). rRNA was removed from the sample using the Ribo-Zero Magnetic Kit (Bacteria) (Illumina, Inc.). The quality of the RNA depletion was then checked using Bioanalyzer (Agilent RNA 6000 PICO Assay, Agilent Technologies, Inc.). The ScriptSeq v2 RNA-Seq Library Preparation Kit (Illumina, Inc.) is then used to make the RNA-Seq library from the Ribo-Zero treated RNA. For each library 1 µg of RNA (rRNA depleted) was used following manufacturer's instructions. The quality of the libraries was evaluated using Bioanalyzer (Agilent Technologies, Inc.).

For the final experiment total RNA was extracted from a total of 20 samples of the growth curve in complex medium (five time points, two technical and two biological replicates) and library sequencing has been carried out at Genomix4Life S.r.l. (Italy) on an Illumina NextSeq500 (single-end sequencing strategy, 1 × 75 bp, ~25 reads/sample).

For real-time PCR (qRT-PCR), total RNA was extracted from the samples of five time points (T4, T6, T8, T10 and T12) for each biological replicate of the growth curve performed in the 19 AA medium, using a RNeasy Mini Kit (Qiagen S.

r.l.), following the manufacturer's instruction. DNA was then removed from the samples using a RNase-free DNase (Qiagen S.r.l.). Overall 10 µl of the extracted RNA was reverse-transcribed using a Superscript II Reverse Transcriptase (Invitrogen) with Random primers (Invitrogen) following the manufacturer's instruction.

**Quantitative real-time PCR (qRT-PCR).** qRT-PCR reactions were performed in a final volume of 10 µl containing 1 µl of a 1:10 dilution of each cDNA, 5 µl of Powrup Sybr Master Mix (Life Technologies) and 1 µM of each primer (Supplementary Table 9). Primers were designed using the Primer3 software[55]. Each sample was spotted in triplicate.

A first experiment using known amounts of DNA of the PhTAC125 strain (1-0.1-0.01-0.001 ng) were performed to obtain a standard curve and calculate the amplification efficiency for each primer pairs (data not shown). *rplM* and *dnaA* genes were used as internal references to normalize mRNA content.

All the reactions were performed on a QuantStudio™ 7 Flex Real-Time PCR System (Applied Biosystems by Life Technologies). Cycling conditions were: hold stage [50 °C for 2′ and 95 °C for 10′], PCR stage [40 cycles of: 95 °C for 30″, 59 °C for 1′, 72 °C for 15″], melt curve stage [95 °C for 15″ 60 °C for 1′, 95 °C for 15′].

**RNA-seq data analysis.** Bowtie 2 (v2.2.3)[56] was used to align raw reads to *P. haloplanktis* TAC125 reference genome (GCA_000026085.1_ASM2608v1). rRNA depletion, strand specificity and gene coverage were evaluated using BEDTools (v2.20.1)[57] and SAMtools (v0.1.19)[58] to verify the library preparation and sequencing performances. Raw read counts were then used to calculate TPM values for each PhTAC125 gene. Clusters of co-regulated genes were identified using the Clust tool[59] using the following parameters: k-means clustering method, tightness weight equal to 0.3 and Q3s outliers threshold equal to 2.0.

Differentially expressed genes between the various contrasts were identified using the R (R Development Core Team, 2012, https://www.r-project.org/) package DeSeq2[60] using default parameters and the following thresholds: adjusted *p* value < 0.01 and log2FC > 0.75 or log2FC < −0.75. The clustering of genes based on their FC was performed using the Pheatmap R package. Visualization of Pearson correlation was performed using "corplot" R package.

**Functional enrichment analysis and regulon identification.** To conduct functional enrichment, each gene whose upstream intergenic region was clustered in one of the three clusters was assigned to a specific functional category using a BLAST[61] search against the COG database[62], with default parameters and considering a hit as significant if *E*-value < 1e−20. The exact binomial test implemented in the R package was used to assess over- and under-represented functional categories against the corresponding genomic background. Available information on PhTAC125 amino acid metabolism regulons were retrieved using RegPrecise database[63,64]. The RegPrecise includes information for 7 PhTAC125 regulons (Supplementary Table 5).

**Motif finding.** Shared, conserved upstream motifs were searched up to 200 bp upstream of the genes belonging to the same regulon. These sequences were retrieved were retrieved from the *P. haloplanktis* TAC125 reference genome and fed into the MEME suite[65] v. 5.1.1. MEME was used in combination with MAST (version 5.0.5)[66] for identifying the most plausible shared motifs upstream of the selected genes. MEME was used setting the following parameters: -nmotifs 5, -minw 6, -maxw 30, -objfun classic, -revcomp, -markov_order 0, -minsites 1, -maxsites 3. All the other parameters were set as default. MAST was used using default parameters. In all cases, only the best scoring motif was considered for further analyses, provided that the search produced a significant result (*e*-value < 0.05). The conservation of identified shared motifs was represented using WebLogo[67].

**Modelling.** The deterministic system was simulated by numerically integrating differential equations using the Matlab built-in function ode45 v. 2019a. To estimate the unknown parameters of the model from experimental data we used a stochastic curve-fitting in-house Matlab software. The algorithm is based on the paper by Cardoso et al.[68] and consists in the combination of the non-linear simplex and the simulated annealing approach to minimize the squared deviation function. To increase the points available for curve-fitting, we used the spline interpolation function implemented in MATLAB on the measured values of nutrients and biomass concentration. The same function was adopted to increase the points available for growth rate estimation during the experiment reported in Fig. 1a.

The codes used to perform the simulations reported in this work and the details about the options of the curve-fitting environment, are available at https://multiauxic.sourceforge.io.

To assess the quality of the fit of the MMM vs. the cybernetic model $R^2$ and AIC were computed. $R^2$ values between experimental data and model predictions were

computed using the built-in *cor.test* function in $R$[69]. AIC values were computed as:

$$AIC = N \times \log\left(\frac{SSE}{N}\right) + 2 \times p.$$

With $p$ being the number of parameters in the models (19 and 9 for cybernetic and MMM models, respectively), $N$ being the number of training cases and SSE being the sum of squared errors for each training set. The interpolated datasets used to compute the fit with the models were also used when computing AIC (i.e., $N = 49$). The AIC values were computed using MATLAB 2019a.

**Growth rates estimation on interpolated growth data**. To increase the points available for curve-fitting, we used the spline interpolation function implemented in MATLAB on the measured values of biomass concentration (OD) obtained from growing PhTAC125 cells on peptone.

**Metabolomic assay and data analysis**. NMR spectra were acquired on (i) cell media to monitor the uptake of the various amino acids by measuring their levels in samples collected at different time points of cell growth; (ii) cell lysates to characterize the intracellular metabolome and its variation over time.

Medium samples were prepared in 5.00 mm NMR tubes by mixing 60 μL of a potassium phosphate buffer (1.5 M $K_2HPO_4$, 100% (v/v) $^2H_2O$, 10 mM sodium trimethylsilyl [2,2,3,3—$^2H_4$] propionate (TMSP), pH 7.4) and 540 μL of sample. Cell lysate samples were prepared in 5.00 NMR tubes by mixing 60 μL of $^2H_2O$ and 540 μL of samples.

Spectral acquisition and processing were performed according to procedures developed at CERM[54,70–74]. All the spectra were recorded using a Bruker 600 MHz spectrometer (Bruker BioSpin) operating at 600.13 MHz proton Larmor frequency and equipped with a 5 mm PATXI $^1H$-$^{13}C$-$^{15}N$ and $^2H$-decoupling probe including a $z$ axis gradient coil, an automatic tuning matching and an automatic and refrigerate sample changer (SampleJet). A BTO 2000 thermocouple served for temperature stabilization at the level of ~0.1 K at the sample. Before measurement, samples were kept for 5 min inside the NMR probe head, for temperature equilibration at 300 K.

For media, NMR spectra were acquired with water peak suppression and (i) one-dimensional (1D) $^1H$ standard NOESY pulse sequence[75] using 128 scans, 65,536 data points, a spectral width of 12,019 Hz, an acquisition time of 2.7 s, a relaxation delay of 4 s and a mixing time of 0.01 s; (ii) two-dimensional (2D) $^1H$-$^{13}C$ heteronuclear single quantum coherence spectroscopy (HSQC) pulse sequence (hsqcetgpsisp2, Bruker). A total of 80 scans were collected using a spectral width of 12,019 for f2 and of 30,178 for f1, f2 and f1 acquisition time of 0.085 sand 0.002 s, respectively, and a relaxation delay of 2 s.

For lysates, NMR spectra were acquired with water peak suppression and (i) 1D $^1H$ standard NOESY pulse sequence using 64 scans, 98,304 data points, a spectral width of 18,028 Hz, an acquisition time of 2.7 s, a relaxation delay of 4 s and a mixing time of 0.01; (ii) 1D $^1H$ Carr–Purcell–Meiboom–Gill sequence using 64 scans[76], 73,728 data points, a spectral width of 12,019 Hz, an acquisition time of 3.07 s and a relaxation delay of 4 s; (iii) 2D $^1H$-$^{13}C$ HSQC pulse sequence (hsqcetgpsisp2, Bruker)[77]. A total of 80 scans were collected using a spectral width of 12,019 for f2 and of 30,178 for f1, f2 and f1 acquisition time of 0.085 sand 0.002 s, respectively, and a relaxation delay of 2 s.

The raw data were multiplied by a 0.3 Hz exponential line broadening before applying Fourier transformation. Transformed spectra were automatically corrected for phase and baseline distortions. All the spectra were then calibrated to the reference signal of TMSP at $\delta$ 0.00 $^1H$ chemical shift (ppm) using TopSpin 3.5 (Bruker BioSpin srl).$^1H$-$^{13}C$ HSQC spectra were also calibrated to the methyl signal of alanine at $\delta$ 19.03 $^{13}C$ chemical shift (ppm).

The signals deriving from each metabolite were assigned using an internal NMR spectral library of pure organic compounds, spiking NMR experiments and literature data. Matching between the present NMR spectra and the NMR spectral library was performed using the AMIX and Assure software (Bruker BioSpin srl). The relative concentrations of the various metabolites were calculated by integrating the corresponding signals in defined spectral range, using a home-made R 3.0.2 script.

**Reporting summary**. Further information on research design is available in the Nature Research Reporting Summary linked to this article.

## Data availability

RNA-seq data that support the findings of this study have been deposited in NCBI SRA archive with the accession codes SAMN12207305 to SAMN12207324. Metabolomics data have been deposited at MetaboLights (https://www.ebi.ac.uk/metabolights/) under code unique identifier MTBLS1699 (www.ebi.ac.uk/metabolights/MTBLS1699).

Databases used in this work: COG (https://www.ncbi.nlm.nih.gov/COG/), KEGG (https://www.genome.jp/kegg/), RegPrecise (http://regprecise.sbpdiscovery.org:8080/WebRegPrecise/).

The authors declare that the other data supporting the findings of this study are available within the paper. No restrictions apply to data availability. The source data underlying Figs. 1a–c, 2a–d, 2b–e, 4a, c, d, 5a, b, S6a, S8a–d, S9a, b, 10a, b, S20 and S21 are provided as a Source Data files.

## Code availability

Code and scripts used to implement the model are available at https://multiauxic.sourceforge.io.

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

## Acknowledgements

P.T. and V.G. acknowledge the support and the use of resources of Instruct-ERIC, a Landmark ESFRI project, and specifically the CERM/CIRMMP Italy Centre. M.F. and E.P. would like to thank Prof. Alessio Mengoni for the help in designing the experiments and his critical evaluation of the work. This project was supported by a PNRA (Programma Nazionale di Ricerca in Antartide) grant (grant PNRA16_00246).

## Author contributions

M.F., E.P. and R.F. initiated the project. M.F. and E.P. designed all the experiments. E.P. performed and/or supervised to all the experiments performed in this work. P.T. and V.G. performed the metabolomic experiments. B.C. and L.C. performed preliminary

transcriptomic experiments. MG contributed during growth and metabolomic experiments and RNA-seq data analysis under the supervision of E.P. and M.F. C.F. contributed to the realization of growth experiments. MF performed/supervised to all the computation reported in this work. M.F. and F.D.P. wrote the theoretical kinetic models and performed the simulations. M.F. wrote the paper. All the authors contributed to the editing of the paper. All the authors have read and approved the final version of the paper.

## Competing interests

The authors declare no competing interests.
