## [Peer Review File · Nature Communications]

Reviewers' comments:

Reviewer #1 (Remarks to the Author):

The paper reports on the observation of the mixed sequential and co-utilization of substrates in a complex environment by a cold-adapted microorganism isolated from Antarctic sea, *Pseudoalteromonas haloplanktis*. In a former study published in BMC Genomics (not cited in the submitted manuscript), the authors integrated literature data into a genome-scale model. The model predicted a wide reorganization of metabolic fluxes along the bacterial growth in a complex medium, as the carbon sources (amino acids) were progressively depleted. The present study has gone a lot further. With dedicated experiments and transcriptomics analyses, the authors were able to identify groups including substrates that can be co-utilized, while substrates from different groups are sequentially utilized. A cybernetic model from the literature was then adapted and fitted to the experimental data. It implements the regulatory constraints that the presence of a substrate induces the expression of the corresponding transport protein, and constraints on the control of the transporter activity and resource allocation. The model fits the data better than classical Monod equations.

The article is clear and well written. Results are original and interesting for the community. Some aspects of the work could have been further developed, see my comments below.

1) Why the authors did not discuss their results in light of those obtained in the previous study published in BMC Genomics? I see these former results obtained with the metabolic fluxes as complementary to the transcriptomics data. It could have been interesting to further inquiry whether the observed gene expression profiles for the different AA regulons are consistent with the predicted fluxes (for data obtained in similar experimental conditions). It could also provide some hints about the regulatory mechanisms. See for instance Chubukov et al Mol Syst Biol 2019, 9:709 as an example of possible analysis to integrate both metabolics and transcriptomics data.

2) Sequential utilization is characterized by the preferred use of the metabolite conferring the highest growth rate/biomass. This is what the authors observe for the different clusters of substrates. The mean biomass reached is higher on C1 substrates, than C2, than C3... However, this comes with a high variability that the authors do not discuss. For instance, Leucine in cluster C2 is preferred over Alanine in Cluster C3, but the growth rate on Leucine is about twice lower than that on Alanine. Do the authors have any explanation for this phenomenon?

3) There is no discussion about the composition of the clusters. Is there some overlap between the catabolic pathways that would explain the co-consumption of substrates belonging to the same cluster? Or regulations of gene expression could be involved?

4) Whether substrates are sequentially used or co-consumed also depends on their own concentration and the environmental growth conditions. For instance, many organisms co-utilize substrates at low concentration while sequential consumption is preferred at higher concentrations. Did they authors try other experimental conditions? If yes, were the clusters identical?

5) Concerning the first model: why is the substrate concentration Φ_i squared in Equation 4?

6) As far as I have understood the second model, the growth rate α is defined as (in LaTeX format): $\alpha = \sum_{j=1}^4 v_j r_j$, which means that there could be some typos in Equation (13). I think that we should have: $\frac{d}{dt}\phi_i = u_i \times r_{\{Ei\}} - \sum_{j=1}^4 v_j r_j \times \phi_i + \beta_i$ or else: $\frac{d}{dt}\phi_i = u_i \times r_{\{Ei\}} - \alpha \times \phi_i + \beta_i$. At least I do not see the reason why Φ_P and Φ_{Si} should appear in the equation. Or am I wrong?

Some minor comments/questions:

a) In the introduction section, readers not familiar with *Pseudoalteromonas haloplanktis* would need some background knowledge (even if it is poor) on the diauxic growth of this microorganism and its regulation.

b) Even though the second model derives from two previously published models, it would be lots easier for the reader to have more details on the underlying assumptions. Is it really necessary to introduce λ_i and P prime, given that they are not used afterwards? They were in the

original literature models, but are not used in this study, at least in a different way, with $\lambda_i=1$ and P prime negligible.

c) Before Equation (10), the summation should be for $j=1$ to 4 (not i)

d) For the second model, β_i should be defined as a constitutive or basal production rate.

e) Which type of R square calculation was used to assess the quality of the fit in Table 2? Given that the two models have different numbers of parameter values, other types of metrics seem more appropriate (e.g. AIC, BIC)

f) There are some English misspellings and typos throughout the text.

Reviewer #2 (Remarks to the Author):

Authors claim that their findings about co-utilization and diauxie together are broadly applicable. However there are a number of issues that are not strong.

Major Concerns

The paper claims about this being general to other bacteria needs to be toned down. Ideally the authors would do these experiments with *E. coli* and other bacteria as well to show further proof of their claims

Figure 1 shows the triauxic curve but it is important to note that the pre-culture is important as is the detail that the medium used was a complex medium with peptone. Under such conditions, things are not representative of the marine ocean environment authors refer to a few sentences earlier. Surely some explanation is needed here.

Figure 1 A needs many more data points to identify a growth rate in the three regions. Otherwise it is really difficult to support the argument for the authors.

Author grow the cells in a more defined AA mixtures (CAA ?) and monitor the decrease in the concentrations corresponding to the three phases. But it is not clear that there is a triauxie here ?

In addition, it is well known that the cybernetic modeling can be used to capture these diauxie/triauxie in cells and so the advance there seems minimal.

Is the distribution in 4E significant ? Authors should show all the ODs of the individual AAs. Can the authors report all the growth rates as in 4F. Also in 4E there is a difference between the growth yield (g/mol) and the growth rate. Authors should measure both in their analysis in order to make proper conclusions. 4E and 4F appears to mix them.

Finally, it is possible that *E. coli* and other model microbes show similar behavior. Unlike C-substrates one would expect co-utilization of AAs as the authors have shown and it is also possible that some AAs are easier to degrade than others and could lead to higher biomass and that is used. But this does not seem to add new insights that I was expecting when I read the abstract.

Reviewer #3 (Remarks to the Author):

Elena Perrin et al. studied by transcriptomics and ¹H NMR metabolomics the time-dependent nutritional behaviour of the Antarctic eubacterium *Pseudoalteromonas haloplanktis*. The heterotrophic bacterium has attracted recent interest due to its anti-biofilm activity by a long-chain fatty aldehyde against *Staphylococci*. On the basis of the genome, the core metabolism is characterized by multiple substrate usages including amino acids, carbohydrates and lipids. This

could reflect the challenges encountered by the bacterium in its natural marine environment requiring rapid adaptation to changing nutrient situations. In this context, some of the authors could already model the metabolic rewiring during growth in a complex medium (BMC Genomics. 2016 Nov 24;17(1):970). Quote? In the current manuscript, the authors carefully determined the transcriptional profiles during the growth of the bacterium in a complex medium displaying a triauxic-like pattern. Functional annotation and clustering provided novel and interesting insights into the dynamics of gene transcription during growth. Concerning metabolism, the patterns suggested that amino acid metabolism is more pronounced in the initial phase of growth whereas carbohydrate metabolism appears to be more important during the late phase. Considering regulation of this phenotype, the authors then focused on transcription factors that did not match the known pattern of other model organisms like *E. coli*. This suggested regulation of metabolic switches by hitherto unknown mechanism that included the differential usage of amino acids during growth. Consequently, the authors determined the time-dependent consumption of amino acids in a defined medium containing amino acids as sole carbon source. Using ¹H NMR analysis, it could be shown that some amino acids were simultaneously metabolized whereas different groups of amino acids are utilized with different dynamics partly explaining the lag-phases of the growth curve. This mixed sequential/diauxic nutrient usage is then simulated by model calculations providing evidence for a highly regulated non-Michaelis-Menten behaviour. This finding is of general interest for the community. On the other hand, further experimental validation is required to justify publication in Nature Communications from my point of view. Most importantly, the metabolic fluxes should be studied in a time-dependent manner by metabolic flux analyses using ¹³C-labelled precursors. This would provide a more direct and unbiased view into the differential usages of amino acids and maybe other substrates. Another point that should be addressed is the novelty of the finding that bacterial co-utilization and sequential uptake of multiple substrates coexist in complex environments. In recent years, similar topologies have been found in e.g. intracellular bacteria growing in the cytosol or vacuoles of host cells by the groups of e.g. Goebel, Eisenreich, Bumann and Abu Kwaik.

In general, a more general view in the reprogramming of the overall metabolic network beyond the usage of amino acid utilization (carbohydrates, lipids, glycerol, citrate cycle intermediates) would benefit the quality of the study.

Minor point: A better documentation of the NMR assignments would be welcome.

Reviewers' comments:

Reviewer #1 (Remarks to the Author):

The paper reports on the observation of the mixed sequential and co-utilization of substrates in a complex environment by a cold-adapted microorganism isolated from Antarctic sea, *Pseudoalteromonas haloplanktis*. In a former study published in BMC Genomics (not cited in the submitted manuscript), the authors integrated literature data into a genome-scale model. The model predicted a wide reorganization of metabolic fluxes along the bacterial growth in a complex medium, as the carbon sources (amino acids) were progressively depleted. The present study has gone a lot further. With dedicated experiments and transcriptomics analyses, the authors were able to identify groups including substrates that can be co-utilized, while substrates from different groups are sequentially utilized. A cybernetic model from the literature was then adapted and fitted to the experimental data. It implements the regulatory constraints that the presence of a substrate induces the expression of the corresponding transport protein, and constraints on the control of the transporter activity and resource allocation. The model fits the data better than classical Monod equations. The article is clear and well written. Results are original and interesting for the community. Some aspects of the work could have been further developed, see my comments below.

Reply: We thank the reviewer for this overall positive evaluation of our work and for considering our results original and interesting for the community. We also agree concerning the possibility to improve some aspects of the work and below we provide a description of the additional analyses performed.

1) Why the authors did not discuss their results in light of those obtained in the previous study published in BMC Genomics? I see these former results obtained with the metabolic fluxes as complementary to the transcriptomics data. It could have been interesting to further inquiry whether the observed gene expression profiles for the different AA regulons are consistent with the predicted fluxes (for data obtained in similar experimental conditions). It could also provide some hints about the regulatory mechanisms. See for instance Chubukov et al Mol Syst Biol 2019, 9:709 as an example of possible analysis to integrate both metabolics and transcriptomics data.

Reply: We thank the reviewer for digging into our previous results published in BMC Genomics and we agree with him that a comparison with the results presented in this manuscript might give further insights to our work. In that previous work, we used multi-step constraint-based metabolic modelling to simulate the growth in a complex medium over several time steps of *Pseudoalteromonas haloplanktis* TAC125 and derived clusters of reactions displaying co-varying metabolic fluxes. We have now compared the measured changes in gene expression for regulons involved in amino acids metabolism during PhTAC125 growth in a complex medium (peptone, the first experiment of this current work) with the predicted fluxes obtained during FBA simulation on a simulated peptone medium from Fondi, Bosi et al. (2016). Results obtained are reported in Supplementary Material S1 (Table S10) and discussed in the text of the new version of the manuscript (see lines 323-329).

2) Sequential utilization is characterized by the preferred use of the metabolite conferring the highest growth rate/biomass. This is what the authors observe for the different clusters of substrates. The mean biomass reached is higher on C1 substrates, than C2, than C3... However, this comes with a high variability that the authors do not discuss. For instance, Leucine in cluster C2 is preferred over Alanine in Cluster C3, but the growth rate on Leucine is about twice lower than that on Alanine. Do the authors have any explanation for this phenomenon?

Reply: Data used to plot figure 4E were gathered from a previous publication of our group (see reference 29 in the main text) where we used Phenotype Microarray (PM) technology to assess (among other things) the metabolic capabilities of PhTAC125 at two different growth temperatures, i.e. 4 and 15 °C. We agree with the observation that, despite the overall trend is pretty clear, there is much variability in the OD values obtained when the carbon sources used in our 19 AA experiment were used as single carbon sources. This might be due to a series of factors that, in our opinion, mostly include i) temperature and ii) cell-to-cell aggregation. In our previous work, PM experiments were conducted at 15°C, whereas growth experiments in this current work were performed at 21°C. Such difference might, in some cases, lead to different growth kinetics and/or biomass yields that, in turn, might account for the variability depicted in Figure 4E. Furthermore, PhTAC125 cells are known to undergo cell-to-cell aggregation in certain nutritional conditions (doi.org/10.1007/s00792-016-0813-2, 10.3389/fcimb.2017.00046 and Perrin et al., *in preparation*). The formation of cellular aggregates impairs the correct quantification of cellular biomass by means of OD quantification and, again, leads to outlier values in Figure 4E. Concerning the difference between Leucine and Alanine growth rates, in our opinion, this can be explained as follows. L-alanine is rapidly metabolized through transamination to pyruvate thanks to the activity of *yfbQ* (PSHAa1323). Conversely, Leucine is commonly degraded through a longer pathway that starts with the activity of *ilvE* and/or *bcd* (PSHAa1270 and PSHAa1167, respectively) and that leads to the production of Acetyl-CoA. As a matter of fact, *ilvE* resulted to be overexpressed in our RNAseq experiment, suggesting that the catabolism of Leucine really occurs through that pathway in the first stages of growth.

Explaining why Alanine is used later than Leucine despite it seems to guarantee a higher growth rate in respect to Leucine, is quite tricky. We think that, promiscuous uptake of nutrients might, at least in this case, explain the incongruencies between growth rate and order of uptake. Indeed, the broad spectrum of some amino acids transporters is well described (doi: 10.1128/JB.184.15.4071-4080.2002, doi.org/10.1016/S0923-2508(01)01197-4) in bacteria. Thus, some amino acids might be up-taken from the medium not as the result of an active cellular control over the most efficient carbon sources but as the results of the broad spectrum activity of a membrane transporter. We have summarized these considerations in the new version of the manuscript (see lines 404-417)

3) There is no discussion about the composition of the clusters. Is there some overlap between the catabolic pathways that would explain the co-consumption of substrates belonging to the same cluster? Or regulations of gene expression could be involved?

Reply: We agree about the lack of a discussion concerning the order of consumption of amino acids in the previous version of the manuscript. We have checked more carefully this point using the amino acid degradation scheme reported in (see reference 24 in the main text). We noticed that the overlap among the catabolic pathways responsible for the assimilation of the amino acids included in the same cluster can partially explain the clustering obtained on the basis of their measured trends in concentration. Indeed, C1 amino acids (arginine, glutamate and glutamine) share part of their catabolic pathways and are all converted to alpha-ketoglutarate before entering the TCA cycle. Two (out of four) amino acids of cluster C2 are converted to Oxaloacetate before entering the TCA cycle. Most of the C3 amino acids (six out of nine amino acids, i.e. Alanine, Serine, Glycine, Threonine, Phenylalanine, Tyrosine) are converted to pyruvate, thus sharing the feature of not being directly catabolized into one of the TCA cycle intermediates. Other two C3 amino acids can be catabolized to form pyruvate and a TCA intermediate (i.e. Succinyl-CoA). The composition of cluster 4 reflects the fact that their usage during PhTAC125 growth is negligible so no degradation pathway is required to explain their inclusion in the same cluster. A schematic representation of this feature is now included in the manuscript (FigureS7 of the new version of the manuscript) and a discussion on the composition of the different clusters at the light of their overlapping catabolic pathways is provided at Lines 418-429.

4) Whether substrates are sequentially used or co-consumed also depends on their own concentration and the environmental growth conditions. For instance, many organisms co-utilize substrates at low concentration while sequential consumption is preferred at higher concentrations. Did they authors try other experimental conditions? If yes, were the clusters identical?

Reply: We thank the reviewer for pointing out this important aspect. This observation prompted us to perform a set of additional experiments aimed at evaluating the effect of higher concentrations of nutrients on the pattern of nutrients assimilation. In the first experiment, we provided the same 19 amino acids at concentrations that were sensibly higher than the one of the original 19 AA experiment. More in detail, we tested nutrients concentration 5 and 10 times higher than the original one and evaluated the effect on the overall assimilation patterns of amino acids. Overall, these two experiments confirmed the results obtained when growing PhTAC125 on the 19 AA medium at a concentration of 0.2 mM in that the amino acids belonging to the same group are simultaneously metabolized by cells but the assimilation of different groups occurs with slightly different dynamics. Concerning the clustering of the amino acids into different groups according to their assimilation kinetics, we observed an overall scenario that resembled the one observed in the 0.2 mM experiment. Cluster 1 (C1) amino acids (Glu, Gln and Arg) are readily and quickly metabolized by PhTAC125, regardless of their initial concentration. Conversely, amino acids belonging to Cluster 4 (C4, Met, Trp and His) are not used by growing PhTAC125 cells, at least in the analysed time frame. This suggests that these latter amino acids are probably assimilated only upon exhaustion of the other 16, as observed in the 0.1 mM 19-AA experiment. In-between these two major clusters, the remaining 13 amino acids are consumed with slightly different rates in all the performed experiments and, as a result, producing slightly different groups during the clustering.

In an additional experiment, we have investigated the effect of nutrients concentration on the “canonical” diauxic shift observed in the original experiment, i.e. the one occurring after the exhaustion of the first set of 16 amino acids and the starting of C4 amino acids assimilation. Specifically, we investigated whether providing these late-metabolized amino acids at higher concentration could influence the timing of this metabolic switch. We provided the three amino acids that were the last to be assimilated in the original experiments (Met, His and Trp) at higher concentration (2 mM) in respect to the concentration of the others present in the same medium (still provided at a concentration of 0.2 mM) and checked whether this influenced their order of uptake in the same 19 AA medium used before. This new experimental condition didn't influence the major split between early metabolized amino acids (Cluster 1 to 3 of the original experiment) and the late-metabolized ones (Cluster 4).

These new data are reported in Supplementary Material (Figure S8 and S9) of the new version of the manuscript. Additionally, the text has been modified to account for these novel findings, see lines 430-443 of the new version of the manuscript).

5) Concerning the first model: why is the substrate concentration Φ_i squared in Equation 4?

The first model was implemented as follows (considering the specific case of a hypothetical substrate A):

This chemical formula accounts for the consumption of nutrient A and the consequent formation of new bacterial biomass (P). The rate of this equation was assumed to follow a Michaelis-Menten kinetics where

$$r = \frac{\beta_A * \phi A}{k_A + \phi A}$$

It follows that the change in time of A concentration will be given by the product of the rate of A and P consumption times the concentrations of A and P:

$$\frac{d\phi A}{dt} = -\frac{\beta_A * \phi A}{k_A + \phi A} * \phi A * \phi P$$

$$\frac{d\phi A}{dt} = -\frac{\beta_A * \phi A^2}{k_A + \phi A} * \phi P$$

We hope this will help in clarifying the formulation of the model as described in the corresponding section. For sake of clarity, we have better explained in the text these steps leading to the equation describing nutrients consumption in our MM kinetic model.

6) As far as I have understood the second model, the growth rate α is defined as (in LaTeX format): $\alpha = \sum_{j=1}^4 v_j r_j$, which means that there could be some typos in Equation (13). I think that we should have: $\frac{d}{dt}\phi_i = u_i \times r_{Ei} - \sum_{j=1}^4 v_j r_j \times \phi_i + \beta_i$ or else: $\frac{d}{dt}\phi_i = u_i \times r_{Ei} - \alpha \times \phi_i + \beta_i$. At least I do not see the reason why ϕ_P and ϕ_{Si} should appear in the equation. Or am I wrong?

Reply: We agree with the reviewer concerning the typos in Equation (13). In the code used to perform the simulation, we actually wrote this equation as:

```
ddy(6) = u1*rE1 -(v1*r1 + v2*r2 + v3*r3 + v4*r4)*y(6)- edr*y(6) +
BasicEnzymeSynthesisRate ;
```

with $y(6)$ referring to the concentration of the enzyme ϕ_i and edr referring to the (physiological) enzyme degradation rate. The other variables have names that match the ones indicated by the reviewer. The first term of the right hand side of this equation accounts for enzyme production, the second term refers to the effect of dilution of the specific enzyme level due to cell growth, the third term reflects the presence of a hypothetical enzyme degradation rate and the final one considers the basic enzyme synthesis rate. We have now corrected the text accordingly (see the new version of Equation 13).

On a side note, other typos that were present in the model formulation were corrected.

Some minor comments/questions:

a) In the introduction section, readers not familiar with *Pseudoalteromonas haloplanktis* would need some background knowledge (even if it is poor) on the diauxic growth of this microorganism and its regulation.

Reply: We agree with this comment of the reviewer. We have introduced two paragraphs in the introduction highlighting what is known about PhTAC125 growth in a nutritionally complex environment. See lines 93-106 of the new version of the manuscript.

b) Even though the second model derives from two previously published models, it would be lots easier for the reader to have more details on the underlying assumptions. Is it really necessary to introduce λ_i and P' , given that they are not used afterwards? They were in the original literature models, but are not used in this study, at least in a different way, with $\lambda_i=1$ and P' negligible.

Reply: We acknowledge that the description of the cybernetic model was rather cryptic in the previous version of the manuscript so we expanded the description of the assumptions behind such an approach (see lines 598-606). Also, as suggested by the reviewer, we have removed the both λ_i and P' from the model formulation.

c) Before Equation (10), the summation should be for $j=1$ to 4 (not i)

Reply: We agree. We have fixed this.

d) For the second model, β_i should be defined as a constitutive or basal production rate.

Reply: We agree. We have added this definition when introducing β_i

e) Which type of R square calculation was used to assess the quality of the fit in Table 2? Given that the two models have different numbers of parameter values, other types of metrics seem more appropriate (e.g. AIC, BIC)

Reply: We thank the reviewer for this suggestion on considering the number of parameters when comparing the two models. Besides the squared Pearson correlation coefficient, we computed AIC metrics to assess the quality of the fit.

Results matched those obtained when computing R^2 , with MMM and cybernetic models showing similar agreement in the fit with biomass values and with the cybernetic model outperforming MMM on compounds consumption prediction in three cases out of four. AIC have been included in the new version of Table 2 (and a brief description on the procedure adopted has been included in Material and Methods, lines 965-976). BIC computation gave similar outcomes and were not included in the revised text.

f) There are some English misspellings and typos throughout the text.

Reply: The text was carefully checked by all the authors. Misspellings and typos were fixed.

Reviewer #2 (Remarks to the Author):

Authors claim that their findings about co-utilization and di-auxie together are broadly applicable. However there are a number of issues that are not strong.

Major Concerns

The paper claims about this being general to other bacteria needs to be toned down. Ideally the authors would do these experiments with *E. coli* and other bacteria as well to show further proof of their claims.

Reply: We thank the reviewer for critically evaluating our work and for this specific comment that gives us the possibility to better clarify our scope. It was not our intention to sustain the claim that our findings are general to other bacteria. On the contrary, we believe that our current knowledge on the possible bacterial strategies for nutrients assimilation on multiple-sources media is biased by the fact that it has been mainly studied in a few model organisms and then generalized to other microorganisms. Our intention here was to characterize a non-model response to nutrients switching and to study the process of bacterial nutrients uptake in experimental conditions that more closely resemble a natural setting, in terms of nutrients heterogeneity, not in terms of their concentration as pointed out below. We think that this overall idea emerges from some sentences of the manuscript: e.g. “we have characterized a non-model response”, “Using a marine heterotrophic bacterium”, “we show that bacterial co-utilization and sequential uptake of multiple substrates can coexist”. In other words, we made all the efforts to stress the fact that the specific feeding strategy that we have been observing in these experiments is one of the possible strategies and that his diffusion inside the microbial kingdom is yet to be determined. As suggested, we have revised the manuscript and corrected all those sentences that could be wrongly interpreted as being general to the microbial world.

Further, in order to get a glimpse on the possible difference in nutrients uptake strategies by different microbes, we have performed the 19-AA experiment using *E. coli* as a test case. Results obtained are detailed in the response to the last point raised by the reviewer and in the new version of the manuscript.

Figure 1 shows the triauxic curve but it is important to note that the pre-culture is important as is the detail that the medium used was a complex medium with peptone. Under such conditions, things are not representative of the marine ocean environment authors refer to a few sentences earlier. Surely some explanation is needed here.

Reply: We agree with the reviewer concerning the necessity to provide more explanation to this part of the text. With “conditions that resemble the ones found in nature” we referred to the availability of many different substrates simultaneously and the necessity to exploit such nutritional condition as quick as possible. We agree with the reviewer that the tested conditions could by no means mirror the ones found in nature concerning the concentration and heterogeneity of nutrients available. We have rephrased the text to make this point clearer, see lines 776-777 of the new version of the manuscript.

Figure 1 A needs many more data points to identify a growth rate in the three regions. Otherwise it is really difficult to support the argument for the authors.

Reply: We understand the point raised here by the reviewer. Since it was not possible to add more data points from the original experiment to Figure 1A, we used an interpolation strategy to get more data points for growth rate calculation. In particular, we used the spline interpolation implemented in MATLAB as explained in “Material and Methods”, section “Modelling”. We then re-computed the growth rates across the time intervals considered. Data obtained were in agreement with the growth rates computed on the original growth curve. The description of this procedure has been included in the new version of the text (Lines 128-131) and the plots showing OD and growth rate estimations based on interpolated have been included in the updated version of Figure S1.)

Author grow the cells in a more defined AA mixtures (CAA ?) and monitor the decrease in the concentrations corresponding to the three phases. But it is not clear that there is a triauxic here ?

Reply: We understand the difficulty of the reviewer in identifying the growth lags in the original version of Figure 4. For this reason, we have now computed and plotted the instantaneous growth rate for each point of the 19-AA experiment growth curve. This plot clearly shows two sudden drops of cellular growth rate during PhTAC125 growth on the 19-AA medium, that results in the two growth arrests observed when measuring OD. These new data are now visible in the revised version of Figure 4. As a side note, the more defined medium the reviewer is referring to is not a casoaminoacid-based medium but an *ad hoc* medium realized for this particular work. Details on how this medium was prepared are provide in Material and Methods section, Lines 809-15.

In addition, it is well known that the cybernetic modeling can be used to capture these diauxie/triauxie in cells and so the advance there seems minimal.

Reply: The reason why we used two different modelling approaches (kinetic Michaelis-Menten vs. cybernetic modelling) relies in the fact that, a priori, we could not rule out the hypothesis that such (light) growth lags could be due to different uptake and usage kinetics of the various amino acids rather than a well-defined control strategy on which amino acid to degrade and use for growth. Since cybernetic modelling framework takes into account the (still) unknown regulatory processes controlling the microorganisms' uptake kinetics, the finding that such model better fits with our data suggests that an active regulatory process is ongoing and influences the order by which amino acids are used by PhTAC125.

Is the distribution in 4E significant? Authors should show all the ODs of the individual AAs. Can the authors report all the growth rates as in 4F. Also in 4E there is a difference between the growth yield (g/mol) and the growth rate. Authors should measure both in their analysis in order to make proper conclusions. 4E and 4F appears to mix them.

Reply: We have carefully reconsidered Figure 4 in the light of the observations made by reviewer #1 and #2. We reckon that the previous structure of Figure 4 was misleading and we would like to point out the following aspects. First, Figure 4E referred to data obtained in another work (Mocali et al. 2017) using Phenotype Microarray and reflected the achievable OD after 167 hours when PhTAC125 cells were grown using each single amino acid as the sole carbon and energy source. To obtain this figure, we retrieved the ODs for all the C1, C2, C3 and C4 amino acids separately and then built the box plot shown in Figure 4E. Similarly, Figure 4F reported growth rates of 4 amino acids from a previous work (Giuliani et al. 2011) when these four amino acids were used as sole C and energy source in a fed-batch growth. Also, in this case, the order of consumption was derived from our metabolomics data. However, for this analysis, to infer the order of consumption, we didn't rely on the cluster of each amino acid but we considered the time in which the concentration of each amino acid reached the value of 0. To avoid any misleading interpretation of Figure 4, we have moved Figure 4 E and F to Supplementary Material S1 (Figure S6). Further, as only one comparison of Figure 4E turned out to be statistically significant, we have toned down the claim on the possible role of achievable growth rates/biomass on the order of consumption of the 19 amino acids (see 406-420 of the new version of the manuscript). We thank the reviewer for pointing out this important aspect that gave us the chance to adjust the focus of the manuscript.

Finally, it is possible that *E. coli* and other model microbes show similar behavior. Unlike C-substrates one would expect co-utilization of AAs as the authors have shown and it is also possible that some AAs are easier to degrade than others and could lead to higher biomass and that is used. But this does not seem to add new insights that I was expecting when I read the abstract.

Reply: We thank the reviewer for pointing out this important aspect. To broaden the scope of our work, we have included additional experiments in our manuscript and, more specifically, we have repeated the same experiments performed with PhTAC125 (those whose results are shown in Figure 4) also for *Escherichia coli*. Details of these new experiments are included in the new version of the manuscript (lines 439-444 and Figure S10 and S11). Overall, we noticed that the behaviour of *E. coli* in a nutritionally rich (but defined) medium differs from the one described for PhTAC125 for the following aspects: i) a large fraction of the amino acids provided in the medium (9 out of 19) are scarcely or not used at all by *E. coli* as carbon/energy sources. These amino acids are the following: Val, Leu, Ile, Met, Asn, Lys, Tyr, His, Phe. ii) This latter amino acid list includes some of those that, on the contrary, are readily utilized by PhTAC125 (e.g. C2 amino acids Leu and Asn). iii) Other amino acids that are not used by PhTAC125 (e.g. Trp) are instead used by *E. coli* in its first growth stages. iv) No growth lags are observed during the growth of *E. coli* on the 19 AA medium, suggesting that no major switches in the uptake of nutrients occur or, at least, that these occurs without influencing the overall cellular reprogramming (as for example seen during classic *E. coli* diauxic growth). Further, while the choice of dividing the amino acids catabolized by PhTAC125 into 4 distinct clusters is suggested by the outcome of an elbow approach, the same analysis conducted over *E. coli* data is more ambiguous and partially supports a separation between 2 main amino acid clusters (catabolized vs. not catabolized). These data have been added to the updated version of the manuscript (Figure S10 and S11).

In our opinion, the phenotypes displayed by the two microorganisms considered here are rather different and we speculate that the specific strategy adopted by PhTAC125 for thriving in a nutritionally heterogeneous medium reflects its adaptation to its natural environment and could provide novel hints on possibly overlooked features of microbial metabolism in non-model organisms and/or in unconventional ecological niches.

Reviewer #3 (Remarks to the Author):

Elena Perrin et al. studied by transcriptomics and 1H NMR metabolomics the time-dependent nutritional behaviour of the Antarctic eubacterium *Pseudoalteromonas haloplanktis*. The heterotrophic bacterium has attracted recent interest due to its anti-biofilm activity by a long-chain fatty aldehyde against Staphylococci. On the basis of the genome, the core metabolism is characterized by multiple substrate usages including amino acids, carbohydrates and lipids. This could reflect the challenges encountered by the bacterium in its natural marine environment requiring rapid adaptation to changing nutrient situations. In this context, some of the authors could already model the metabolic rewiring during growth in a complex medium (BMC Genomics. 2016 Nov 24;17(1):970). Quote? In the current manuscript, the authors carefully determined the transcriptional profiles during the growth of the bacterium in a complex medium displaying a triauxic-like

pattern. Functional annotation and clustering provided novel and interesting insights into the dynamics of gene transcription during growth. Concerning metabolism, the patterns suggested that amino acid metabolism is more pronounced in the initial phase of growth whereas carbohydrate metabolism appears to be more important during the late phase. Considering regulation of this phenotype, the authors then focused on transcription factors that did not match the known pattern of other model organisms like *E. coli*. This suggested regulation of metabolic switches by hitherto unknown mechanism that included the differential usage of amino acids during growth. Consequently, the authors determined the time-dependent consumption of amino acids in a defined medium containing amino acids as sole carbon source. Using ¹H NMR analysis, it could be shown that some amino acids were simultaneously metabolized whereas different groups of amino acids are utilized with different dynamics partly explaining the lag-phases of the growth curve. This mixed sequential/diauxic nutrient usage is then simulated by model calculations providing evidence for a highly regulated non-Michaelis-Menten behaviour. This finding is of general interest for the community. On the other hand, further experimental validation is required to justify publication in *Nature Communications* from my point of view.

Reply: We thank the reviewer for the careful analysis of our manuscript and for providing positive feedbacks to our work. We understand the need of further experimental validation to fully convince the reviewer about the relevance and robustness of our work. Below we address the points raised by her/him.

Most importantly, the metabolic fluxes should be studied in a time-dependent manner by metabolic flux analyses using ¹³C-labelled precursors. This would provide a more direct and unbiased view into the differential usages of amino acids and maybe other substrates.

Reply: As suggested, we have carried out additional experiments to study in a time-dependent manner the fate of the metabolized amino acids inside the PhTAC125 cells. More in detail, we have repeated the experiment described in the earlier version of the manuscript about the consumption of the 19 amino acids (results shown in Figure 4) using ¹³C labelled amino acids to track the fate of such compounds inside the cell. We have selected one representative for each of the 4 clusters of metabolized amino acids (Figure 4B), namely Glu, Asp, Ala and Met, and used their ¹³C uniformly-labelled version to perform four different experiments, using one different ¹³C amino acid each time. For each growth experiment, two precultures (as described in the text for the original experiment) were used, while the final growth experiments were performed in quadruplicate. Cell growth was monitored measuring the OD₆₀₀ every hour. At four time points, early and late exponential growth (3 and 6 hours) and early and late stationary phase (8 hours and 30 minutes, and 24 hours), one of the 4 replicates was analysed. A scheme of the adopted experimental strategy is now included in Supplementary Material (Figure S12). This analysis allowed to study the metabolic fluxes in a time-dependent fashion and provided hints on the fate of the metabolized amino acids inside the cell. In particular, we were able to show that Ala is mostly used to feed all the important pathways inside the cell, namely gluconeogenesis, pentose phosphate pathway, nucleotide precursors metabolism and (partly) TCA cycle. Similarly, Asp seems to be converted to PEP (a central precursor for gluconeogenesis and pentose phosphate pathway) and, more importantly, utilized for nucleotide (AXP and GXP) biosynthesis. Further, Asp is also used to feed the TCA cycle (principally through the formation of fumarate) and for the biosynthesis of NAD precursors. Following its uptake, Glu seems to be readily redirected towards the TCA. Interestingly, Glu seems also to be used as a substrate for NAD (and precursors) biosynthesis. Our data also suggests a central role for Asp in PhTAC125 metabolism as ¹³C atoms of this amino acid are found almost everywhere in the central metabolic network. We also confirmed that Met is incorporated into the PhTAC125 metabolic network at a later stage of its growth, in that no identified compounds were labelled with Met ¹³C at T3 (after 3 hours). Met labelled carbons appear to be included into homocysteine starting from T6 (6 hours). After 24 hours, we found Met labelled carbons on a compound whose NMR pattern corresponds to that of methylamine. Interestingly, in a previous work, we had characterized PhTAC125 as a methylamine producer and also showed that adding Met to the growth medium was pivotal to allow PhTAC125 to produce this compound (and inhibit the growth of human opportunistic pathogens). This additional body of data is described at pages 13-16 of the new version of the manuscript.

Another point that should be addressed is the novelty of the finding that bacterial co-utilization and sequential uptake of multiple substrates coexist in complex environments. In recent years, similar topologies have been found in e.g. intracellular bacteria growing in the cytosol or vacuoles of host cells by the groups of e.g. Goebel, Eisenreich, Bumann and Abu Kwaik.

Reply: We thank the reviewer for suggesting the comparison with the important contributions from the authors above. We had missed this body of literature in the previous version of the manuscript. We have added additional text to highlight the analogies with what is currently known for intracellular bacteria (see lines 57-60 and Lines 782-786). We speculate that these phenotypes (different and plastic carbon assimilation strategies) might be common to organisms that are used to face nutritionally poor environments.

In general, a more general view in the reprogramming of the overall metabolic network beyond the usage of amino acid utilization (carbohydrates, lipids, glycerol, citrate cycle intermediates) would benefit the quality of the study.

Reply: We agree with the reviewer concerning the possibility to broaden the analysis beyond the study of amino acids degradation by looking at changes in the internal concentration of specific metabolites. For this reason, we analysed the changes in relative concentration of a panel of intracellular metabolites. The whole intracellular metabolome was evaluated through untargeted ¹H NMR across four time-points, i.e. early and late exponential growth (3 and 6 hours) and early and late stationary phase (8 hours and 30 minutes, 24 hours). The spectra of twenty-six metabolites could be unambiguously resolved and quantified. These metabolites were representative of 6 major metabolic modules, i.e. amino acids

metabolism, purine and pyrimidine metabolism, sugars, aminosugars metabolism, nucleotide precursors and TCA. A by-product compound (formate) was also monitored.

We show that the relative concentration of purine and pyrimidines (and ribose) intermediates decreases in time, compatibly with the decrease in growth rate induced by the exhaustion of carbon sources. Similarly, TCA intermediates (fumarate) halve their relative concentration during PhTAC125 growth. Conversely, sugars (e.g. glucose) and amino sugars (e.g. UDP-NAG) increase their cellular concentration in the latest growth stages, reaching values that are up to three times the initial ones. This latter finding can be interpreted in the light of two considerations: first, we have already shown that carbohydrates metabolic genes increase their expression at the final stages of PhTAC125 growth on peptones, suggesting a switch from an amino acid-based metabolism to a carbohydrates-based one. Second, amino sugars might have a role in PhTAC125 cell aggregation given that i) they are known to be involved in cell-cell contacts (<https://doi.org/10.1159/000369583>) and ii) that PhTAC125 was shown to produce a biofilm that incorporates aminosugars to survive in a strategy to survive in poor nutrient conditions. Likely, the presence of a lower availability of nutrients induces a greater production of biofilm since the biofilm matrix can improve the capture of nutrients [doi.org/10.1016/j.micres.2018.09.010]. We have added these new data as additional files (Figure S19 and S20) and discussed these results in the updated version of the manuscript (Lines 538-563)

Minor point: A better documentation of the NMR assignments would be welcome.

Reply: As suggested, we have added tables (Figures S5, S14-S20) with the ¹H NMR assignments for each of analysed metabolites in both growing media and cell lysates.

REVIEWERS' COMMENTS:

Reviewer #1 (Remarks to the Author):

The authors carefully addressed all the points made previously. The new results and analyses are convincing. I think the paper has improved significantly and recommend it for publication.

Minor comments (some misspellings and one comment that the authors may want to take into account in the final version to improve clarity)

- The revised manuscript could have been a bit more explicit on the FBA analysis (pp. 15-16 of Supplementary Material S1). Why using two different approaches? What is "nutritional" MOMA? How were calculated the Fisher's Z transformation average of Pearson correlation coefficients shown in Table S10? Along time and within each gene regulon?
- Line 606 of the manuscript: problem in the definition of i
- Line 825: missing word in "To confirm the order of AA WE USED Schatz salts"
- Supplementary Material S1 : missing word in title "Relationship BETWEEN the order of consumption"

Reviewer #3 (Remarks to the Author):

The authors have addressed all of my concerns quite carefully on the basis of additional experiments providing further evidence for the conclusions.

Concerning the references for the intracellular bacteria. The person who initiated this line of thinking is Werner Goebel. The authors have now added original work most of which was stimulated by his ideas. To take this into account, the authors should add 1-2 of his review articles, maybe

To Eat and to Be Eaten: Mutual Metabolic Adaptations of Immune Cells and Intracellular Bacterial Pathogens upon Infection.

Eisenreich W, Rudel T, Heesemann J, Goebel W.

Front Cell Infect Microbiol. 2017 Jul 13;7:316. doi: 10.3389/fcimb.2017.00316. eCollection 2017. Review.

and/or

Metabolic Adaptations of Intracellular Bacterial Pathogens and their Mammalian Host Cells during Infection ("Pathometabolism").

Eisenreich W, Heesemann J, Rudel T, Goebel W.

Microbiol Spectr. 2015 Jun;3(3). doi: 10.1128/microbiolspec.MBP-0002-2014.

and/or

Carbon metabolism of intracellular bacterial pathogens and possible links to virulence.

Eisenreich W, Dandekar T, Heesemann J, Goebel W.

Nat Rev Microbiol. 2010 Jun;8(6):401-12. doi: 10.1038/nrmicro2351. Epub 2010 May 10. Review.

Minor mistake:

line 801: Coxiella burnetii delete the d

REVIEWERS' COMMENTS:

Reviewer #1 (Remarks to the Author):

The authors carefully addressed all the points made previously. The new results and analyses are convincing. I think the paper has improved significantly and recommend it for publication.

Minor comments (some misspellings and one comment that the authors may want to take into account in the final version to improve clarity)

- The revised manuscript could have been a bit more explicit on the FBA analysis (pp. 15-16 of Supplementary Material S1). Why using two different approaches? What is "nutritional" MOMA? How were calculated the Fisher's Z transformation average of Pearson correlation coefficients shown in Table S10? Along time and within each gene regulon?

Reply: We thank the reviewer for pointing out this minor issues. The referee is correct, Fisher's Z transformation average of Pearson correlation were computed along time and within each gene regulon. We have also added more details concerning the use of nutritional-MOMA.

- Line 606 of the manuscript: problem in the definition of i

Reply: Fixed

- Line 825: missing word in "To confirm the order of AA WE USED Schatz salts"

Reply: Fixed

- Supplementary Material S1 : missing word in title "Relationship BETWEEN the order of consumption"

Reply: Fixed

Reviewer #3 (Remarks to the Author):

The authors have addressed all of my concerns quite carefully on the basis of additional experiments providing further evidence for the conclusions.

Concerning the references for the intracellular bacteria. The person who initiated this line of thinking is Werner Goebel. The authors have now added original work most of which was stimulated by his ideas. To take this into account, the authors should add 1-2 of his review articles, maybe

To Eat and to Be Eaten: Mutual Metabolic Adaptations of Immune Cells and Intracellular Bacterial Pathogens upon Infection.

Eisenreich W, Rudel T, Heesemann J, Goebel W.

Front Cell Infect Microbiol. 2017 Jul 13;7:316. doi: 10.3389/fcimb.2017.00316. eCollection 2017. Review.

and/or

Metabolic Adaptations of Intracellular Bacterial Pathogens and their Mammalian Host Cells during Infection ("Pathometabolism").

Eisenreich W, Heesemann J, Rudel T, Goebel W.

Microbiol Spectr. 2015 Jun;3(3). doi: 10.1128/microbiolspec.MBP-0002-2014.

and/or

Carbon metabolism of intracellular bacterial pathogens and possible links to virulence.

Eisenreich W, Dandekar T, Heesemann J, Goebel W.

Nat Rev Microbiol. 2010 Jun;8(6):401-12. doi: 10.1038/nrmicro2351. Epub 2010 May 10.
Review.

Reply: we inserted the citation to the most recent review of Werner Goebel and its group "To Eat and to Be Eaten: Mutual Metabolic Adaptations of Immune Cells and Intracellular Bacterial Pathogens upon Infection."

Minor mistake:

line 801: Coxiella burnetiid delete the d

Reply: Fixed